# A case study of percutaneous epidural stimulation to enable motor control in two men after spinal cord injury

Ashraf S. Gorgey [1,2] ✉, Robert Trainer[3], Tommy W. Sutor [1], Jacob A. Goldsmith[1], Ahmed Alazzam[1], Lance L. Goetz[1,2], Denise Lester[3] & Timothy D. Lavis[1,2]

Two persons with chronic motor complete spinal cord injury (SCI) were implanted with percutaneous spinal cord epidural stimulation (SCES) leads to enable motor control below the injury level (NCT04782947). Through a period of temporary followed by permanent SCES implantation, spinal mapping was conducted primarily to optimize configurations enabling volitional control of movement and training of standing and stepping as a secondary outcome. In both participants, SCES enabled voluntary increased muscle activation and movement below the injury and decreased assistance during exoskeleton-assisted walking. After permanent implantation, both participants voluntarily modulated induced torques but not always in the intended directions. In one participant, percutaneous SCES enabled motor control below the injury one-day following temporary implantation as confirmed by electromyography. The same participant achieved independent standing with minimal upper extremity self-balance assistance, independent stepping in parallel bars and overground ambulation with a walker. SCES via percutaneous leads holds promise for enhancing rehabilitation and enabling motor functions for people with SCI.

Spinal cord epidural stimulation (SCES) may modulate spinal cord neural networks to enhance multiple functions after motor complete spinal cord injury (SCI)[1]. When combined with physical training, SCES has enabled restoration or volitional enhancement of overground standing and stepping[2–5]. Prior human trials have relied primarily on conducting surgical laminectomies to implant a 16-electrode mounted paddle[3–6]. While these have yielded promising results, paddle implantation may not be suitable for all individuals depending on the level of injury, spinal hardware fusion and scar tissue formation. Alternatively, SCES using percutaneous leads does not necessitate laminectomy for implantation and allows immediate hospital discharge post-implantation[7–10]. This may offer people ineligible to receive a paddle a chance to reap the potential benefits of SCES[7]. SCES with percutaneous leads has been used in motor control studies and as a treatment

for spasticity in persons with motor complete SCI[8,9] as well as enabled voluntary knee extension in persons with incomplete[10] or motor complete SCI[11]. However, no reports exist detailing the use of percutaneous SCES to enable overground standing or stepping, or related rehabilitation activities, specifically in persons with motor complete SCI.

This case report describes early results of two persons with clinically sensory-motor complete and a motor complete SCI enrolled in a clinical trial investigating the effects of percutaneous SCES for motor function (NCT04782947; IDE# G190003). Further details are provided in a timeline of study procedures (Fig. 1). In this report, enabling motor control is defined as using SCES to augment supraspinal intent integrated with proprioceptive input to yield compound motor functions such as standing or volitional movements[1].

[1]Spinal Cord Injury and Disorders Center, Hunter Holmes McGuire VAMC, 1201 Broad Rock Boulevard, Richmond, VA 23249, USA. [2]Virginia Commonwealth University, Department of Physical Medicine & Rehabilitation, Richmond, VA 23298, USA. [3]Physical Medicine and Rehabilitation, Hunter Holmes McGuire VAMC, 1201 Broad Rock Boulevard, Richmond, VA 23249, USA. ✉e-mail: ashraf.gorgey@va.gov

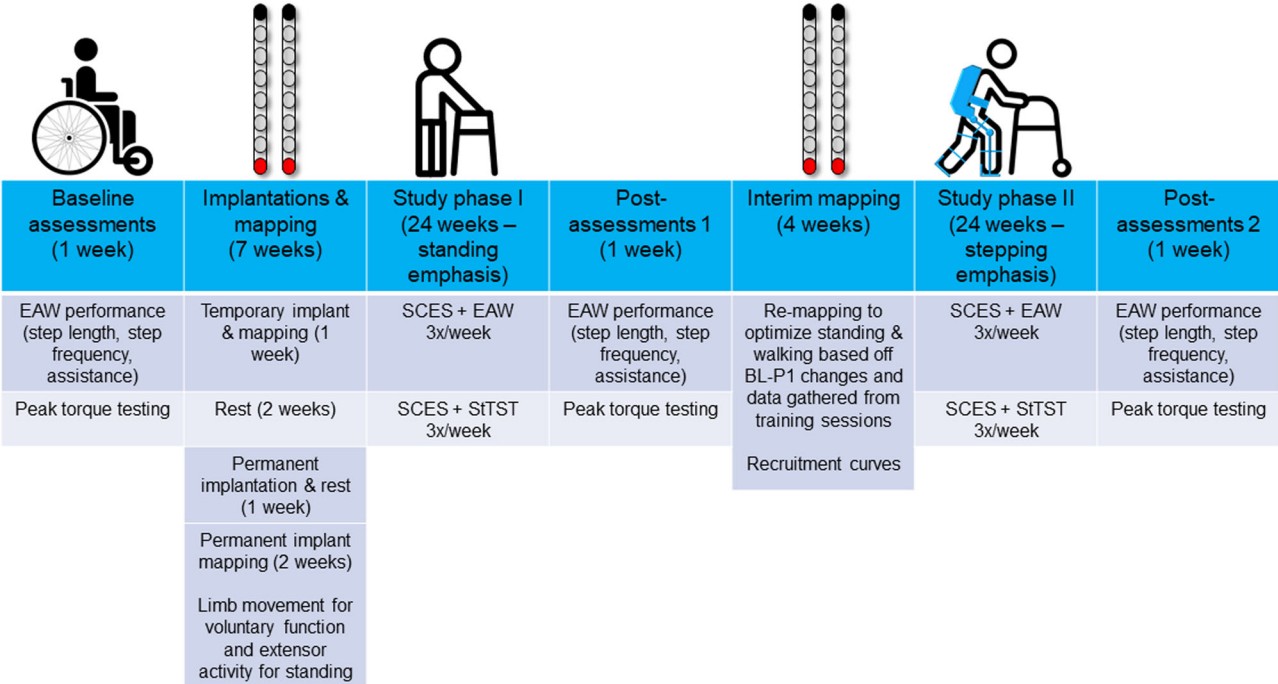

**Fig. 1 | Study timeline.** Timeline of study phases for NCT04782947. After baseline outcome measures are assessed, temporary implantation occurs followed by 5 days of mapping. After removal of the temporary implant, participants are given 2 weeks rest and then receive the permanent implant. Permanent implant mapping occurs for 2 weeks, focusing mainly on voluntary limb movement and extensor activity to facilitate standing. Following this, phase I of the study begins, consisting of 24 weeks training 3 days per week. Each training day consists of one hour of spinal cord epidural stimulation (SCES) combined with exoskeleton-assisted walking (EAW), followed by one hour of SCES combined with standing task-specific training (StTST). After 1 week of re-assessing outcome measures (Post-assessments 1), 4 weeks of re-mapping is conducted to optimize standing, and walking function for the next phases of the study. Following this, phase II of the study begins, consisting of 24 weeks of the same training as phase I. In the final week of the study, outcome measures are re-assessed a final time (post-assessments 2).

We report enabling of various voluntary movements, and enhancement of muscle activity and physical performance during exoskeleton-assisted walking (EAW) in both participants. Additionally, one participant achieved restoration of overground standing. stepping in parallel bars and overground ambulation with a standard walker.

## Results
### Participant characteristics
Two men with clinically motor complete traumatic SCI (C8; 6 years post-injury [ID#: 0772] and T11; 9 years post-injury [ID#: 0773]), American Spinal Injury Association Impairment Scale [AIS] A and B, respectively) participated in a trial approved by the Hunter Holmes McGuire Veteran Affairs Medical Center ethical IRB committee. AIS exam sheets for both participants and magnetic resonance images for 0772 are provided in Supplementary Figs. 1–3. This is an interim reporting analysis of the clinical trial that was approved by the data and safety monitoring board (DSMB).

### Temporary and permanent implantation
Both participants underwent temporary SCES implantation (5 days) followed by permanent implantation 4 weeks later (Supplementary Figs. 4–5). Two leads were implanted to provide multiple cathodal and anodal configuration options either combined or separately to enhance functional outcomes. Temporary implantation was conducted in accordance with regulations for use of percutaneous SCES electrodes, and to allow a trial mapping period to ensure mapping could be successfully conducted with permanent implants. Procedures for placement of the temporary SCES leads lasted ~45 min. With participants prone, the epidural space was accessed with a 14-gauge needle and loss of resistance technique, and a percutaneous SCES system (Intellis Epidural Stimulator, Medtronic, Minneapolis, USA) with two 8-electrode leads was implanted utilizing fluoroscopic guidance (Supplementary Figs. 4–5). For temporary implantation, the leads were navigated in the epidural space, taped and glued to the skin, and then connected to an external neurostimulator (Intellis, Medtronic 97725) that was secured externally to the lateral side of the trunk for 5 days. On the 5th day of the temporary implantation, both participants underwent x-ray fluoroscopy to assess lead migration before de-implantation. Permanent implantation procedures were similar to those for the temporary implant, except an implantable stimulator (Intellis, Medtronic 97715) was secured under the skin near the implanted leads. Permanent implantation was followed with 3–4 weeks of instruction not to perform strenuous physical activities without immobilization, during which spinal mapping took place. The permanent implant for 0772 was placed from the T11-L1 vertebrae to cover the lumbosacral enlargement (Supplementary Fig. 4). Participant 0773 had spinal hardware and excessive scar tissue that necessitated staggered positioning at T12-L1 (left lead) and L1-L2 (right lead). Supplementary Figures 4 and 5 provide images and further details of temporary and permanent lead implantation. Twelve weeks following permanent implantation, modest lead migration occurred in both 0772 and 0773 participants; however, within the acceptable limits[12,13].

### Spinal mapping
Spinal mapping was carried out to identify optimal configurations (cathodal-anodal electrode arrangements and stimulation parameters) to enable multiple functions and movements without inducing unwanted activity[14,15]. Spinal mapping was conducted daily after temporary (1 week) and permanent (2 weeks) implantation, as well after the first 6 months of the study in an interim mapping phase (4 weeks). The emphasis of the first six months of the study was to achieve overground standing, while the emphasis of the second six months was to achieve step-like activity to lead to overground ambulation.

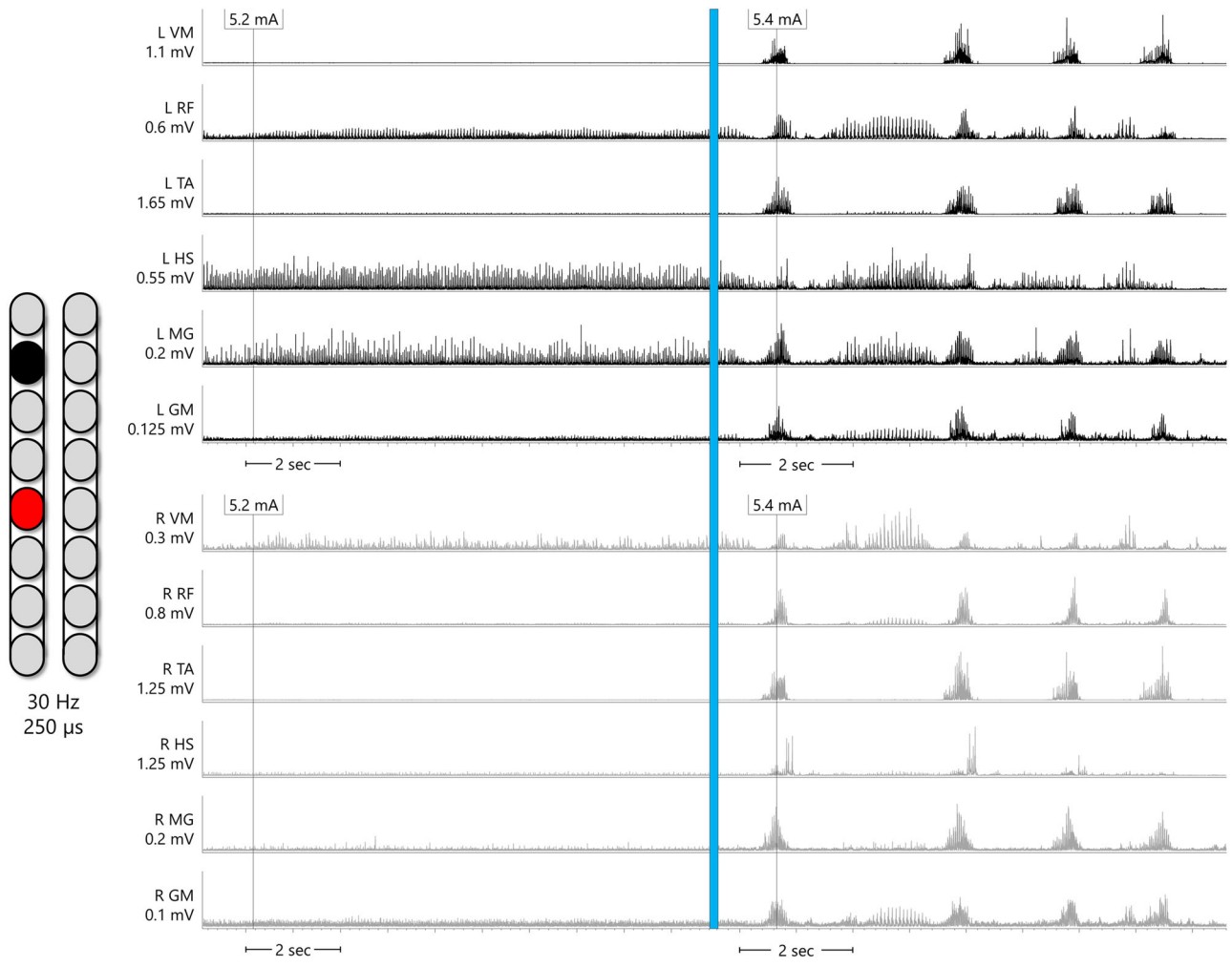

**Fig. 2 | Supine rhythmic EMGs.** Example electromyograms (EMGs) of left leg muscles (black traces) and right leg muscles (gray traces) while SCES is delivered in supine at various amplitudes to participant 0773, using the configuration shown on the left of the figure (black is the cathode and the red is the anode). In the left column of EMGs, SCES delivered at 5.2 mA induces little tonic activity or no activity at all. In the right column of EMGs, the same SCES configuration is delivered at 5.4 mA. Upon ramping of stimulation to 5.4 mA, a single burst of activity across all muscles is induced, followed by periods of relaxation and then subsequent bursts of similar shape. Each burst of muscle activity across all muscles resulted in sudden, bilateral knee, hip and ankle flexion, followed by the legs returning to resting on the table during periods of relaxation. EMGs presented are rectified and bandpass filtered at 10–990 Hz. L left, R right, VM vastus medialis, RF rectus femoris, TA tibialis anterior, HS hamstring, MG medial gastrocnemius, GM gluteus medius, mV millivolts, mA milliamps, Hz hertz, μs microseconds, sec seconds.

Thus, the initial temporary and permanent mapping periods prior to the first 6 months emphasized voluntary limb movement and extensor activity to facilitate overground standing. In the interim mapping phase, these original configurations were refined. Configurations which yielded tonic extensor activity in supine were refined to enable standing with the participant upright, first fully supported in a standing frame. Further refining was done by having participants attempt a sit-to-stand in an exoskeleton (EksoNR, Ekso Bionics, CA, USA) in "squat" mode, a manufacturer setting which provides assistive torque at the knees and hips sufficient to complete a sit-to-stand only if the user volitionally contributes to the movement. Lastly, configurations were refined with participants standing in parallel bars. In addition to refinement of configurations for standing, during the interim mapping phase, further mapping was conducted for rhythmic locomotor activity to facilitate stepping and overground ambulation. Simple configurations (one cathode and one anode) mapped using existing methods[16] yielded rhythmic bursting in some lower extremity muscles in both participants, exemplary data of 0773 were presented (Fig. 2) in a similar fashion shown previously with percutaneous SCES[15]. Results yielded by each mapping process are described in the subsequent sections.

## Voluntary motor activity

Participant 0772 did not elicit intentional voluntary movements in supine or side lying positions. His legs moved in response to different SCES configurations following temporary implantation. On day 1 of temporary mapping, in the presence of percutaneous SCES, 0773 voluntarily flexed his right hip from a side lying position on command (Supplementary Fig. 6). When in supine, the same configuration induced bilateral, tonic activity in the legs; however, attempts to flex his right hip resulted in increased right leg electromyography (EMG) activity with concurrent inhibition of left leg activity (Fig. 3). On the second day, he flexed his left leg and dorsiflexed his left ankle on command.

The participants' ability to volitionally generate isometric knee extension torque in a seated position was tested using the configurations that induced or enabled knee extension in supine. Testing took place 12 weeks (0772) or 5 weeks (0773) after permanent implantation. Detailed peak voluntary torque methods are described in the Supplementary methods. A representative torque trace is shown in Supplementary Fig. 7 and a summary of torque testing results are in Supplementary Table 1. Overall, both participants were able to modulate torque induced by SCES, though not always in the intended

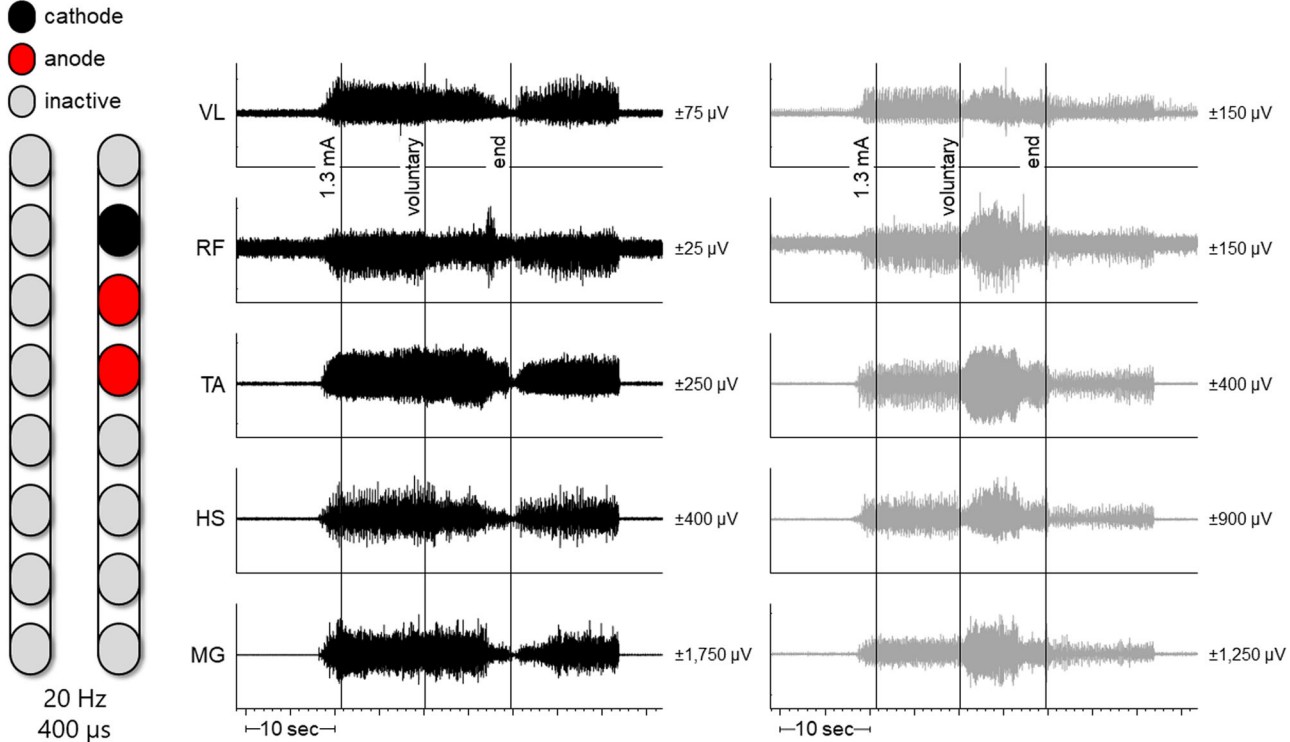

**Fig. 3 | Configuration for 0773 resulting in right leg flexion.** Left leg muscles are depicted in black and right leg muscles in gray. Both plots occurred simultaneously but have been separated to better visualize different effects of voluntary effort between legs. At 1.3 mA, bilateral muscle activity was induced. However, when the participant volitionally attempted to flex his right leg, muscle activity decreased in some left leg muscles (VL, RF, and HS) while increasing or bursting in right leg muscles (RF, TA, HS, and MG). Hz hertz (stimulation frequency), μs microseconds, L left, R right, VL vastus lateralis, RF rectus femoris, TA tibialis anterior, HS hamstrings, MG medial gastrocnemius, sec seconds, μv microvolts.

direction. However, with SCES on, volitional torque time integral (TTI; Nm/sec) was greater compared to volitional attempts without SCES or compared to TTI induced by SCES (see Supplementary Table 1).

### Standing ability

In the first phase of the study, with SCES on, 0772 did not need to use his upper extremities for balance self-assistance in a standing position. In contrast when SCES was off, he needed to hold the parallel bars and use his upper extremities to self-assist balance. However, 0772 required maximal knee assistance and moderate hip assistance from study staff to stand in parallel bars whether SCES was on or off, and this persisted throughout the whole study. Despite this, during the interim mapping process (following the first phase of the study) when configurations for standing were refined, 0772 demonstrated volitional trunk and hip control while in a semi-standing position, supported in the standing frame; further details are provided in Fig. 4. After the interim mapping, he could also voluntarily sit-to-stand in the exoskeleton in "squat" mode with SCES on, but not with SCES off (Fig. 5). In the presence of SCES, participant 0773 achieved overground standing in parallel bars, with no external assistance from the research team and use of the upper extremities only for self-balance assistance (Fig. 6) in approximately week 6 after permanent implantation. In contrast to the other participant, during interim mapping 0773 was not able to execute a voluntary sit-to-stand maneuver in the exoskeleton, yet his overground standing ability remained unchanged.

### Exoskeleton-assisted walking and overground stepping

Using the aforementioned mapping procedures in the interim period, configurations for rhythmic activity did not yield locomotor-like muscle activity in supine position in that the bursting was synchronous across all muscles in both legs (i.e. lack of a reciprocal relationship between antagonistic muscles or left-right alternations; Fig. 2). Yet when these configurations were tried in combination with EAW, both participants showed an improvement in EAW performance compared to the first phase of the study when configurations for limb movement were used during EAW sessions (Fig. 7). Both participants also showed increased overall EMG activity during EAW; exemplary data for 0773 are presented (Fig. 8). With the same SCES configuration that improved EAW performance, 0773 was able to stand and step in a set of parallel bars and perform overground walking with a walker for 16 steps. This stepping was executed with some self-assistance from the upper extremities with no external assistance from any research staff and could not be replicated with SCES off.

### Discussion

Percutaneous SCES enabled motor control in two persons with chronic motor complete SCI. Both participants could also volitionally modulate induced flexion or extension torques on verbal instruction in a seated position. One participant (0773) initiated hip flexion, knee flexion, and ankle dorsiflexion in a supine lying position following temporary and permanent implantation. The other participant (0772) did not show an immediate ability to volitionally generate movement or EMG activity; however, he was able to voluntarily modulate SCES-induced torque after the permanent implantation, and following interim mapping, demonstrated volitional hip and trunk control when supported in a semi-standing position. Standing configurations in 0772 may have resulted in trunk muscles modulation via inter-neuronal connections with lower lumbosacral segments, which resulted in enhanced trunk control as seen in Fig. 4. Full trunk control has recently been restored following 1 day of target SCES application to T12 dorsal nerve roots in three persons with complete sensorimotor paralysis[17].

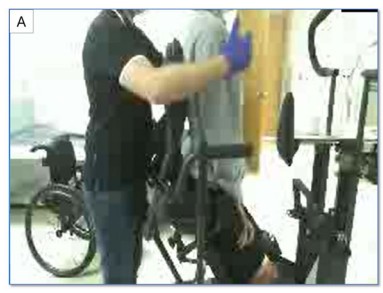
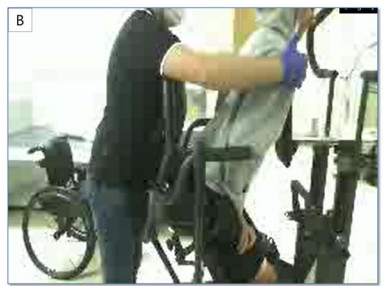
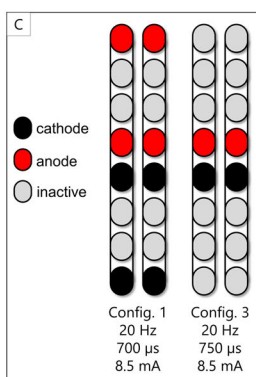
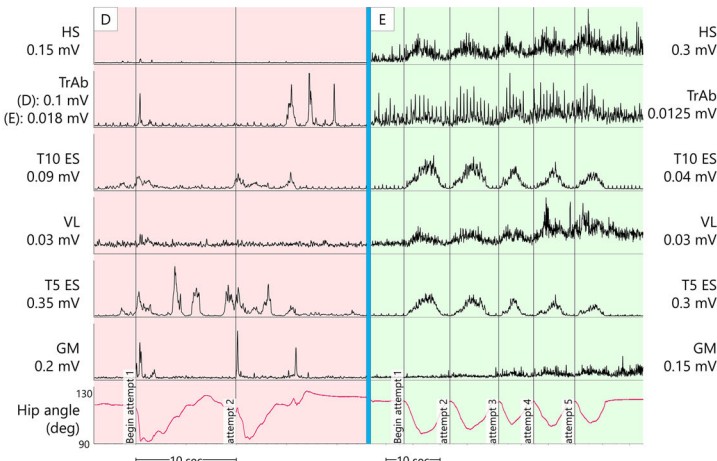
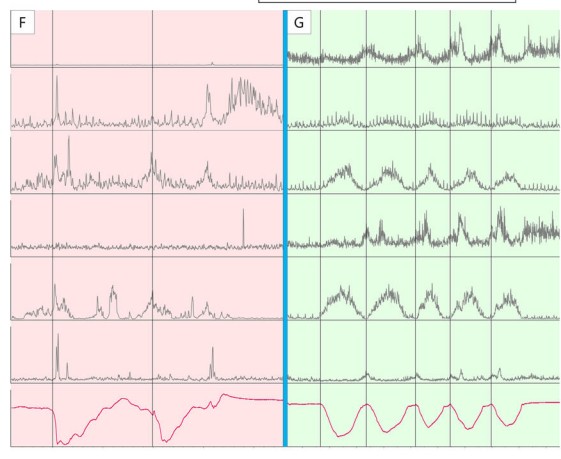

**Fig. 4 | Voluntary hip flexion and extension with trunk control (participant 0772) from a semi-standing position in a standing frame.** Panels **A** and **B** show the start and end of the range of motion, respectively, the participant attempts without self-assistance from the arms or external assistance. A study team member is seen guarding the participant for safety. **C** SCES configurations and stimulation parameters. **D–G** root mean square EMG activation of trunk (T5 ES and T10 ES: erector spinae muscles at T5 or T10 vertebral levels respectively; TrAb: transverse abdominis muscles) and leg muscles. Trunk EMGs had an additional comb filter applied to remove SCES artifact[27], while ECG artifact is still visible. Hip angle is derived from an electrogoniometer on the participant's hip. Panels **D** and **F** show left and right-side muscles, respectively, with the hip angle trace replicated in both panels, while the participant attempts the maneuver with SCES off. Bursts seen in ES and gluteal muscles (GM) are likely reflexive, as they occur synchronously with the rapid decline in hip angle as the participant was unable to control his descent.

Following this, activity in T5 ES coincides with the participant using arm self-assistance to return to the starting position. Panels **E** and **G** show left and right muscles, respectively, with the hip angle trace replicated in both panels, while the participant attempts the maneuver with SCES on. The hip angle shows more controlled descent into flexion and ascent into extension, accomplished without self-assistance or external assistance. EMG activity shown corresponds to hip angle changes – at the lowest hip angle, the participant had the greatest need for trunk stability, and more activity is seen from trunk muscles. Quadriceps and hamstrings increase activity as the participant leans forward, aiding in control of the descent and subsequent ascent. Note that for the left TrAb, y-axis scales are different between panels **D** and **E**, in order to better illustrate change in EMG activity during the maneuver with SCES on. HS hamstrings, VL vastus lateralis, GM gluteus medius, sec seconds, config configuration, Hz Hertz, µs microseconds, mA milliamps.

Both participants demonstrated improved standing ability via execution of a volitional sit-to-stand with exoskeleton assistance (0772) or unassisted standing in parallel bars (0773). It is possible that the muscle activity during these activities (Figs. 5 and 6, respectively) resulted from differences in proprioceptive input with volitional attempt when transitioning from a sitting to weight-bearing position, as has been shown previously[3]. It is worth noted that the effect of proprioceptive input alone was not tested. However, this is unlikely for 0772, as the muscle activity and subsequent movement generated did not occur without volitional effort from the participant (Fig. 5). Without SCES, 0772 could not initiate standing into an upright position even with exoskeleton assistance. Additionally, his overground standing improved in that he no longer needed his upper extremities to self-assist balance.

In participant 0773, we cannot rule out that his standing ability is related to load receptor augmentation, rather than him enabling volitional motor control, that resulted in amplification of extensor tone to sustain standing (Fig. 6). Previous work examined the effects of volitional effort to the contribution of standing motor pattern[18,19]. Mechanistically, SCES may access the spinal circuitry controlling the standing posture and enabled independent standing following just

2 weeks of training in a T3 complete person with SCI via potentially activating spared neural connections below the injury level[18]. In participant 0773, transitioning from sit to stand may have resulted in neuromodulation of the activation pattern necessary to facilitate standing. In clinically sensory-motor complete participants with SCI, SCES neuromodulated proprioceptive afferent inputs during transitioning from sit-to-stand activity[19]. The authors suggested that sensory supraspinal projection is not required, but rather sensory information projected to spinal cord circuitry is required to achieve full weight-bearing standing[19].

Both participants showed increased muscle activity during EAW concurrent with enhanced EAW performance, indicated by faster walking speeds with reduced exoskeleton-provided swing phase assistance. The increased muscle activity manifested mainly as increased EMG amplitudes during the stance phase of gait as opposed to more typical patterns of alternating firing of flexors and extensors during the swing phase. The atypical EMG firing observed during EAW (without SCES) is driven primarily by the exoskeleton-provided flexion and extension pattern[20]. It is clear based on Fig. 8 that the addition of SCES overrides this pattern in the stance phase, but not entirely in the

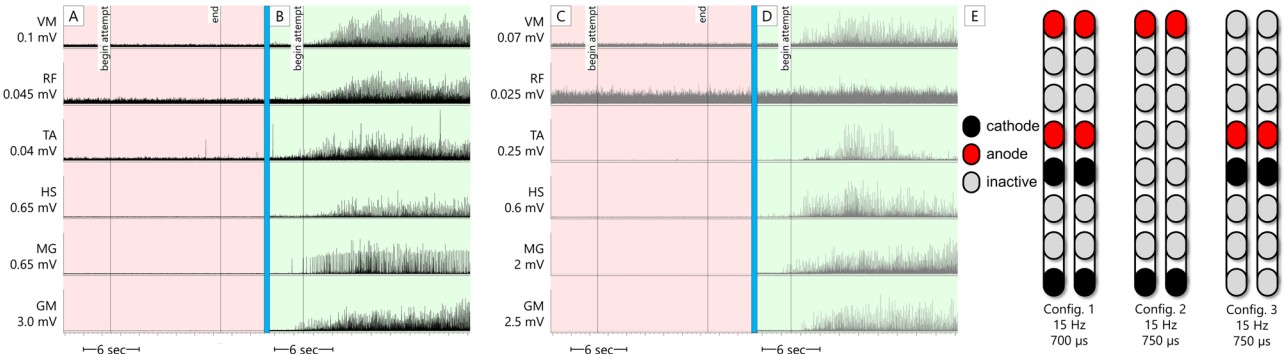

**Fig. 5 | 0772 Exoskeleton sit-to-stands.** Electromyograms (EMGs) of lower extremity muscles of 0772 during active attempts to complete a sit-to-stand maneuver with the exoskeleton in "Squat" mode. In this mode, the exoskeleton will not passively complete a sit-to-stand for the user; rather, assistive torque is provided at the knees and hips which can enable the user to complete a sit-to-stand only if they are able to volitionally contribute to the movement. The assistance level was set to "very high", which is the highest of four levels of manufacturer-determined assistance. **A**, **C** left (black traces) and right (gray traces) leg muscles, respectively, during an active attempt to complete a sit-to-stand with SCES off. 0772 was unable

to generate any muscle activity, and consequently, could not complete the sit-to-stand. **B**, **D** left (black traces) and right (gray traces) leg muscles, respectively, during an active attempt to complete a sit-to-stand with SCES on at 4.2 mA using configurations shown in **E**. As the muscle activity increased, 0772 was successfully able to complete the sit-to-stand. EMGs presented are rectified and band-pass filtered at 10–990 Hz. VM vastus medialis, RF rectus femoris, TA tibialis anterior, HS hamstrings, MG medial gastrocnemius, GM gluteus medius, sec seconds, Hz Hertz, μs microseconds, config configuration.

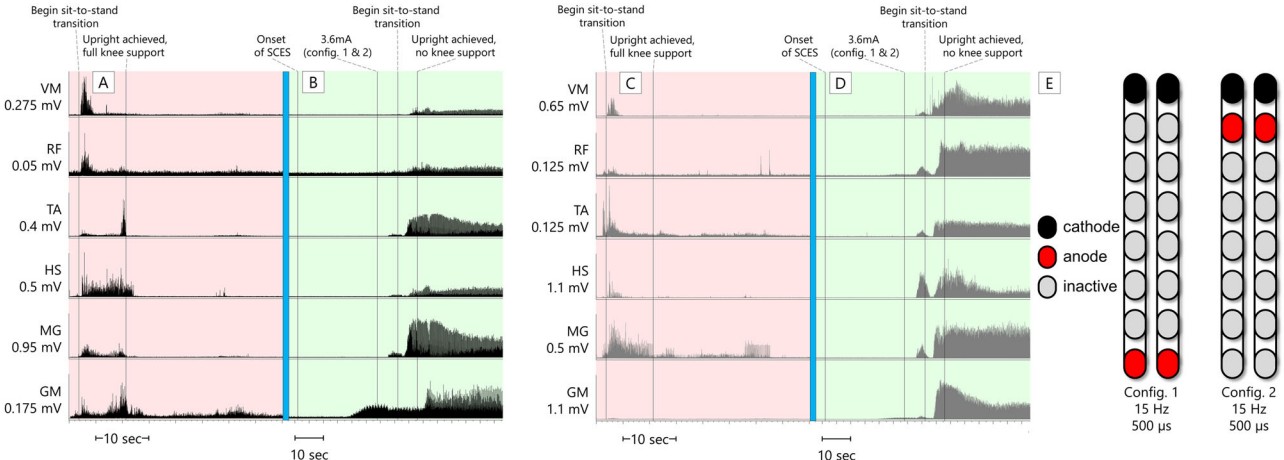

**Fig. 6 | 0773 overground standing.** Electromyograms (EMGs) of lower extremity muscles of 0773 during overground standing in parallel bars. **A**, **C** left (black traces) and right (gray traces) leg muscles, respectively, during upright, overground standing with SCES off. 0773 was able to use his upper extremities to self-assist himself into an upright position with the knees blocked by a study team member, and once upright, required full support at the knees to maintain the upright position. EMG activity during the sit-to-stand transition was a brief period of spasticity – once upright, muscle activity largely ceased. **B**, **D** left (black traces) and right (gray traces) leg muscles, respectively, during upright, overground standing with SCES on using configurations shown in **E**. From the onset of SCES, the amplitude was

gradually ramped up on one configuration at a time before both configurations were set at an amplitude of 3.6 mA. After using upper extremities to self-assist into an upright position with the knees blocked by a study team member, the study team member was able to fully let go of the knees. The participant needed no support to maintain standing aside from balance self-assist with the upper extremities. EMGs presented are rectified and band-pass filtered at 10–990 Hz. VM vastus medialis, RF rectus femoris, TA tibialis anterior, HS hamstrings, MG medial gastrocnemius, GM gluteus medius, sec seconds, Hz hertz, μs microseconds, config configuration.

swing phase. This possibly may be due to the fact that SCES reorganizes the interneuronal firing to attenuate such atypical EMG firing observed during EAW without SCES[1]. However, participant 0773 progressed to unassisted overground stepping with the use of SCES and a standard walker. The stepping occurred without apparent multi-joint flexion which was seen when the configuration was used in supine. It is possible that the participant's extensor tone, which was often stronger when in an upright position (see Fig. 6, panels **A** and **C**) overpowered the flexion seen in supine. The torque at the knees provided by the exoskeleton was strong enough to break this tone during EAW, but without the exoskeleton, he seemed unable to volitionally override this tone.

There are several possible reasons for the discordant results between participants. The first explanation may be differing injury

levels, injury severity (AIS A or AIS B), or differences in preserved muscle mass. Previous studies have also shown position-dependent effects of SCES[15], which may explain why effects of knee extension configurations found in supine did not transfer to seated torque testing. Pursuant to this point, the extensive cortical reorganization of motor areas which occurs after SCI could have affected the precision with which participants could attempt to execute various movements[21,22]. Additionally, the toque data presented in Supplementary Fig 7 and Table 1 showed that torque modulation is physiologically trivial, and it is difficult to know whether it is caused by compensatory movements of the trunk and upper body, which can occur even when the participant is properly strapped on the ergometer, or by activation of key lower limb muscles. However, the torque data for both participants indicated that they

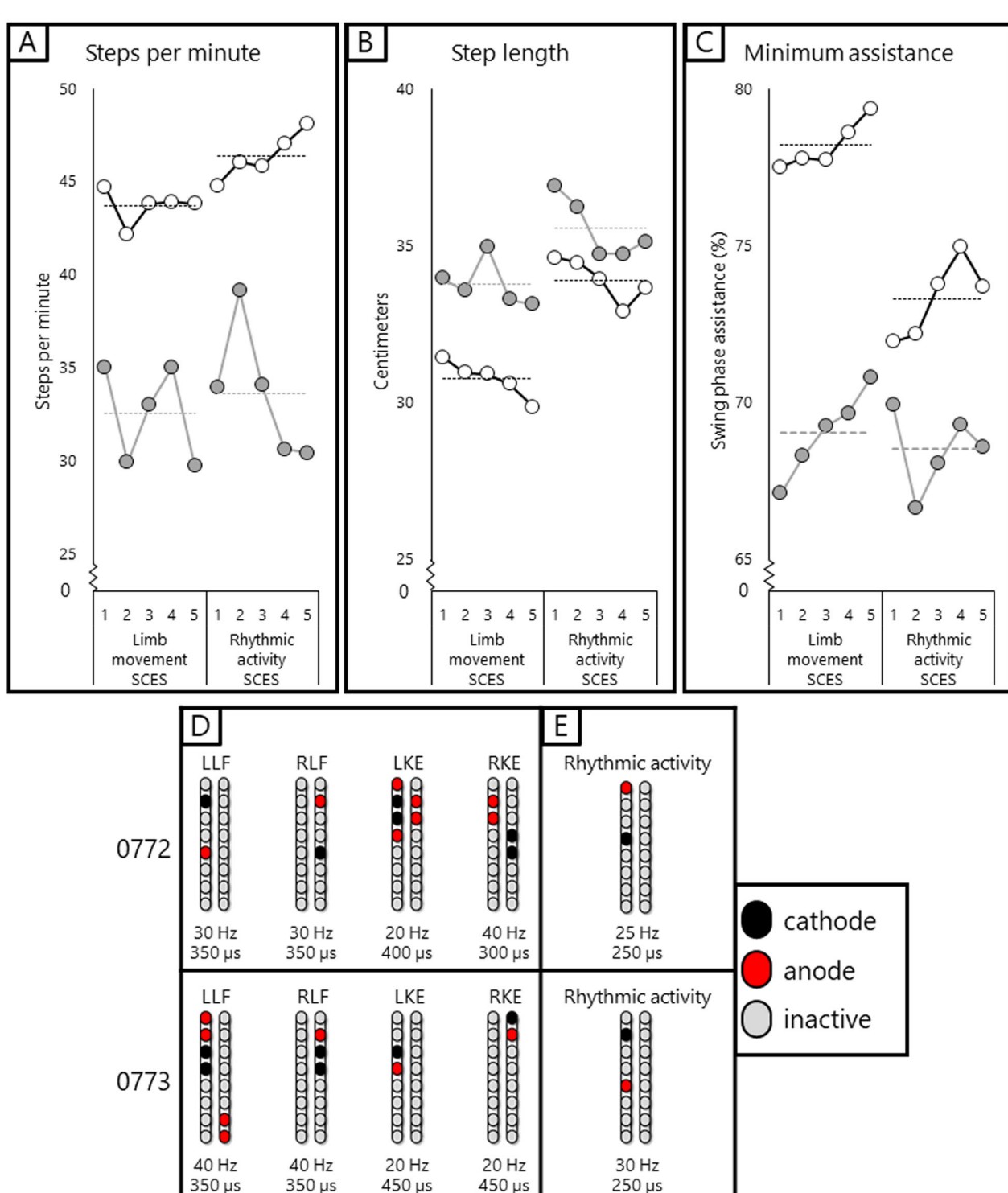

**Exoskeleton-assisted walking enhancement**

were capable of modulating SCES induced torque more than without SCES.

These factors notwithstanding, it is likely that SCES (afferent excitation) combined with the participants' intention to move (supraspinal control) enabled movement in varied contexts for both participants[15,16]. The use of percutaneous leads may increase the accessibility of SCES-enabled motor control for persons with SCI. One

of the participants in this case series was ineligible for a paddle implantation due to scarring and hardware at the injury site, necessitating placement of the percutaneous leads at a location not exclusively covering the L1-S2 spinal segments (Supplementary Fig. 5). Pursuant to location of the electrodes, migration of percutaneous leads in the caudal direction is possible following implantation[23]. A recent study in 91 individuals showed that within 20 days of

**Fig. 7 | EAW enhancement figure.** Data points showing changes in exoskeleton-assisted walking (EAW) performance for two participants walking in "Adaptive" mode, a manufacturer setting which adjusts the amount of assistance provided to the user, and thus can result in variability in certain parameters of gait.
**A–C** individual dots represent the average of 300 consecutive steps across 5 separate EAW training sessions for steps per minute, step length, or minimum exoskeleton-provided swing phase assistance (measured as a percentage of the maximum possible assistance the exoskeleton can provide), respectively. Dotted lines represent the average of each 5-session segment. The "Limb movement SCES" segment of each graph show the individual session data (dots and solid lines) and 5-session average (dotted lines) of the last five EAW sessions using four concurrent SCES programs which facilitated flexion or extension movements for each

individual leg (configurations shown in panel **D**). These data are from the last five sessions of a 24-week period of EAW using these configurations. After mapping for supine rhythmic locomotor activity (configurations shown in panel **E**), improvements are seen in all outcomes for both participants across the first five EAW sessions using this new, single configuration. Increases in steps per minute and step length indicate a faster walking speed, and changes in minimum assistance indicate that the participants achieved said improvements while the exoskeleton provided less assistive torque at the knees and hips during the swing phase of gait. SCES spinal cord epidural stimulation, LLF left leg flexion, RLF right leg flexion, LKE left knee extension, RKE, right knee extension, Hz hertz, μs microseconds. Source data are provided as a Source Data File.

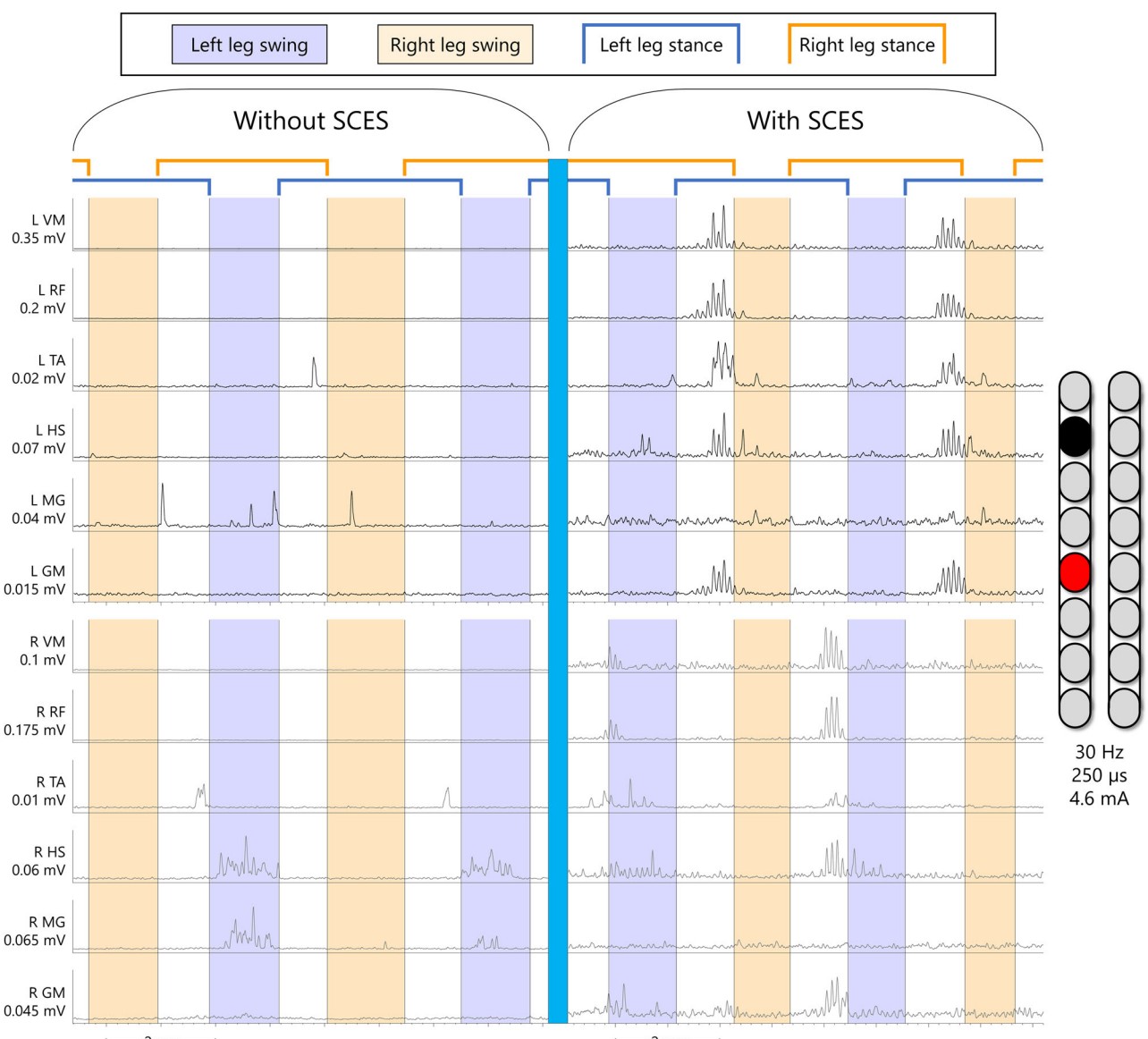

**Fig. 8 | EAW enhancement EMG figure.** Example electromyograms of one participant (0773) during exoskeleton-assisted walking (EAW) without or with spinal cord epidural stimulation (SCES). Left leg muscles are shown in black, right leg muscles are shown in gray. Without SCES, very little muscle activity occurs through the gait cycle, mainly in the right hamstring and medial gastrocnemius muscles, which occurred during the swing phase of the left leg. With SCES on, using the same configuration that resulted in rhythmic bursting in supine as seen in Fig. 1, but at a lower stimulation amplitude, muscle activity is greatly enhanced in nearly all leg muscles. With SCES, the largest bursts in the left leg muscles occur simultaneously,

immediately preceding the right leg swing phase. The largest bursts in the right leg muscles occur in a similar fashion, in that the bursts occur simultaneously through all right leg muscles immediately preceding or slightly overlapping with the beginning of the left leg swing phase. EMGs presented are band-pass filtered at 10–990 Hz, then smoothed with a root-mean square envelope with a moving window of 100 samples. SCES spinal cord epidural stimulation, L left, R right, VM vastus medialis, RF rectus femoris, TA tibialis anterior, HS hamstring, MG medial gastrocnemius, GM gluteus medius, mV millivolts, mA milliamps, Hz hertz, μs microseconds, sec seconds.

implantation, 88.5% of implanted leads had migrated caudally with a mean distance of 1.2 cm (antero-posterior view) and 1.7 cm (lateral view)[24]. In anticipation of this migration, the leads were initially implanted in a more rostral location, and the leads were sutured to the interspinous ligament. Despite taking these measures, caudal migration occurred in both participants in this report to varying extents. However in 0773, migration following permanent implantation was <1.7 cm and within the provisionally acceptable limits[24]. Also, medio-lateral shifting was observed, with part of the contacts crossing midline, which may influence side-specific facilitation of SCES[25]. This may also explain why the use of a single lead for some configurations in this study yielded bilateral muscle activation, given the more medial placement of the leads[23]. However, both participants demonstrated varying degrees of motor control, with one participant – whose leads were placed at lower anatomical segments – demonstrating volitional movements in the paralyzed limbs on the first day of temporary implantation. The potential versatility of SCES via percutaneous leads to enable motor control after SCI deserves further exploration. Finally, we should point out that COVID-19 restrictions have imposed difficulties assessing kinematic data and limited the use of EMG in clinical settings.

Currently, our registered clinical trial is scheduled to recruit 20 participants. Recruiting, implanting, and training more participants will need additional time. Furthermore, our research center is working hard within the COVID-19 precautions to ensure reasonable flow without exposing research staff or patients to additional unnecessary risks. We are reporting results on our first two participants as an interim report to demonstrate the feasibility of our research protocol and to ensure timely dissemination within the SCI community.

In summary, the current report shows percutaneous SCES may enable motor control leading to functional improvements not possible with SCES off. Relatively simple mapping procedures and configurations were used to enhance performance of rehabilitation and training sessions. With these same configurations, some functional benefits manifested without the need for extensive training, such as standing with minimal use of upper extremity and stepping with a walker in one participant. It is possible that the severity of the injury, preserved muscle mass, and the accuracy and position specificity of the spinal mapping may have played a role in the findings of the current report. Further work is needed to refine mapping and training protocols to ensure optimization of this enabled motor control, and to make results more consistent across participants. Future research should investigate the efficacy of percutaneous SCES in combination with physical rehabilitation for restoration of motor control in large cohorts of persons with chronic SCI.

## Methods
### Subjects
Two men with clinically motor complete traumatic SCI (C8; 6 years post-injury [ID#: 0772] and T11; 9 years post-injury [ID#: 0773]), American Spinal Injury Association Impairment Scale [AIS] A and B, respectively) participated in a trial approved by the Hunter Holmes McGuire Veteran Affairs Medical Center ethical IRB committee. AIS exam sheets for both participants and magnetic resonance images for 0772 are provided in Supplementary Figs. 1–3. The participants were recruited from the Hunter Holmes McGuire VA Spinal Cord Dysfunction registry as part of a pilot clinical randomized controlled trial ((https://clinicaltrials.gov/ct2/show/NCT04782947; IDE# G190003) studying the effects of exoskeletal assisted walking (EAW) and spinal cord epidural stimulation (SCES) on volitional motor recovery and other health-related aspects after SCI. This included 3 weeks of measurements (baseline, post-interventions 1 and 2), 2 weeks for implantation and 12 months for training. Study timeline and different phases for NCT04782947 trial were provided in Fig. 1. Participants consented that their data will be used for research publication as a part of the

listed clinical trial. Data safety monitoring board (DSMB) allowed interim reporting analysis for this clinical trial.

### Inclusion and exclusion criteria
Participants may be included if they are between 18 and 60 years old, male or female, with traumatic motor complete SCI and level of injury of between C5 to T10, as determined the International Standards for Neurological Classification of SCI (ISNCSCI) exam. Participants' knee extensors must respond to standard surface NMES procedures (frequency: 30 Hz; pulse duration: 450 µs and amplitude of the current: up to 200 mA) to ensure intact neural circuitry below the level of SCI. All participants will undergo ISNCSCI examination for neurological level and function and only those with American Spinal Injury Classification (AIS A and B; i.e. motor deficit below the level of injury) will be included. The inclusion of AIS A and B ensures we can simply detect any additional volitional control below the motor level of injury. A caregiver or companion must be available to assist subjects who require assistance. We have chosen to set the age limit to 60 years as the upper limit of the study, because persons above 60 years old are likely to have cardiovascular problems that can prevent engagement in strenuous physical activity for 12 months. A written clearance by the medical doctor to ensure that the participant was safely able to engage in the program. Women on contraceptives may be included in the study.

Participants may be excluded from the current trial if any of the following pre-existing medical conditions are present: (1) Diagnosis of neurological injury other than SCI, including cauda equina or distal conus injuries resulting in limb or sacral areflexia; (2) Unhealed fracture in either lower or upper extremities; (3) Severe scoliosis, hip knee range of motion (ROM) or flexion knee contractures preventing positioning in an exoskeleton or plantarflexion contracture >20°. (4) Untreated or uncontrolled hypertension defined as high resting blood pressure >140/90 mmHg and severe orthostatic hypotension (drop greater than 20 mmHg compared to resting supine blood pressure) or incapable to maintain a sitting or EAW standing posture; (5) Other medical conditions including cardiovascular disease, uncontrolled type II diabetes mellitus, uncontrolled hypertension, and those on insulin, or symptomatic urinary tract infection; (6) Unable to fit in the device for any reason; (7) Taking anti-coagulants or anti-platelet agents, including aspirin if unable to be off this medication for medical reasons; (8) Implanted pacemakers and/or implanted defibrillator devices; (9) DXA T-Score <−2.5 standard deviations of the total body. Total hip bone mineral density (BMD) T-scores <−3.5 and knee BMD scores of less than 0.6 g/cm²; (10) Functional upper and lower extremity ROM, strength, spasticity and skin integrity are also assessed prior to enrollment in the program. The Modified Ashworth Scale will be used to ensure safety of the participants prior to engagement in the rehabilitation program. Participants with severe spasticity or limited ROM may be excluded from the trial. This is based on the Ekso® manufacturer's recommendations; (11) Untreatable severe spasticity judged to be contraindicated by the site Physician; (12) Pressure ulcer of the trunk, pelvic area, or lower extremities of grade 3 or more; (13) Psychopathology documentation in the medical record or history that may conflict with study objectives; (14) Any medical condition that, in the judgment of the principal investigator or medical provider, precludes safe participation in the study and/or increases the risk of infection. (15) Allergic reaction to minocycline and rifampin; and (16) Pregnant women and women who will be involved and become pregnant during the course of the study will be excluded as well. A blood pregnancy test will be conducted to rule out any pregnancy before the study. The test will be repeated every month during the course of the study. The blood samples will be sent to the pathology lab for analysis.

### Magnetic resonance imaging
Prior to enrollment in the trial, participants were asked to conduct magnetic resonance image (MRI; T2 Turbo Spin Echo with long band

width; SIEMENS 1.5T) with the following scanning sequence (slice thickness: 3 mm, TR: 9350, TE: 102; flip angle: 150) for pre-screening purpose. MRI was conducted to verify the injury site and determine the extent of injury (Supplementary Fig. 3). Magnetic resonance images for 0773 participant were not available due to the interfering spinal hardware in the region of SCI.

### Interventions

**Implantation of epidural stimulation.** This is a 2-step process where temporary implantation precedes permanent implantation when indicated. The SCES system (Intellis Epidural Stimulator, Medtronic, Minneapolis, USA) was used to electrically stimulate the lumbosacral enlargement. During temporary implantation, two 8-electrode lead arrays were implanted utilizing fluoroscopic guidance over spinal cord segments T10-L2. Prior to both temporary and permanent procedures Hibiclens® (chlorhexidine) soap skin cleanser and Bactroban® (mupirocin) 2% ointment is given for 7 days prior to the procedure to reduce bacterial colonization of participant's skin and nares. An anesthesia preoperative evaluation was performed, and consent obtained prior to entrance into the operating room. We have split the process of implantation into temporary and permanent to ensure appropriate placement of the leads. Additionally, possible unanticipated medical events may emerge following temporary implantation that may lead to withdrawal from the study or to be considered as a screen failure or the patient may deny participation because feeling of discomfort or pain.

**Temporary implantation.** Participants were scheduled to perform the temporary implantation. After placing the participant in a prone position, the participant was implanted in a minor procedure room under fluoroscopy guidance. A nurse certified in sedation established IV access, place standard ASA monitors including noninvasive blood pressure every 5 min, pulse oximetry, continuous EKG, and end tidal $CO_2$ from a nasal cannula. Antibiotics (typically Ancef 2–3 grams or Clindamycin 600–900 mg) was used at the time of implantation. Through provided 14-gauge epidural needles using x-ray guidance and loss of resistance technique the epidural space was accessed. Next, the leads were navigated in the epidural space and the configuration (i.e. stimulation parameters), in the presence of Medtronic representative, necessary to evoke motor potentials was tested as indicated by visible motor contractions of the paralyzed muscles[26]. The leads were steered to the left and right of midline on live AP fluoroscopy confirming posterior epidural placement on lateral fluoroscopy. After rudimentary testing on the fluoroscopy table, lead position was optimized after proper motor stimulation was confirmed. The needles were removed with the leads remaining in the epidural space[24]. The electrodes were taped and glued to the skin during the procedure to lessen the chance of lead migration. Temporary implantation was conducted to successfully ensure activation of the lumbosacral segments prior to conducting permanent implantation. Furthermore, the participant was given time to consider whether he/she wants to proceed with permanent implantation. Participants were asked to use chlorhexidine scrub to the low back and Bactroban applied to the nares twice a day for the 7 days prior to permanent implantation.

**Permanent implantation.** Fourteen days following temporary implantation, two 8-electrode arrays of Vectris lead were implanted in an operating room (see the listed details about the Intellis System). The trial was undertaken in a sterile environment with the presence of a representative. Phase 1 of the permanent implantation is identical to the temporary trial described but with the anesthesia provided by an anesthesiologist. An IV line was established and standard ASA monitors was placed including noninvasive blood pressure every 5 minutes, pulse oximetry, continuous EKG, and end tidal $CO_2$ from a nasal

cannula. Antibiotics (typically Cefazolin 2-3 grams or Clindamycin 600-900 mg) were used at the time of implantation. Through provided 14-gauge epidural needles using x-ray guidance and loss of resistance technique the epidural space will be accessed. The leads were navigated in the epidural space and correct configurations (i.e. stimulation parameters) necessary to evoke motor potentials were re-tested. An incision in the participant's lower back or their buttock was performed to place the pulse generator in a pocket of tissue between the muscles and the skin. The leads were anchored to ligament and/or fascia with non-absorbable sutures. The leads were then threaded under the skin to a pocket, they were then connected to a Medtronic Intellis battery and impedances checked after hemostasis and irrigation applied. Following hemostasis, the wound was closed in 2–3 layers, derma-bond, occlusive dressing and tape were placed over the wound. An abdominal binder was provided for participant's comfort.

Prescription for pain medicine was only for 3 days as this seems to be the most painful part as the single incision heals. Recovery after implant was complete at the 7–10 days mark when bandages were removed. Participants were examined 3 times in the first month for wound check (1 week), dressing change (1 week and 2 week) and reprogramming (week 4). Permanent implantation was followed with 3–4 weeks of instruction not to perform strenuous physical activities without immobilization, during which spinal mapping took place in week 4. Rostro-caudal and medio-lateral migration of the leads were evaluated following both temporary (5 days post-impanation) and permanent implantation.

**The Intellis system has two main sets of components.**
1- External components for Intellis
- Model 97715/97716 Wireless External Neurostimulator
  The Medtronic Model 97715/97716 Wireless External Neurostimulator (ENS) is part of a neurostimulator system used for intraoperative testing during lead placement and for trial stimulation outside of the operating room. The Medtronic Model 97715/97716 Wireless External Neurostimulator is a disposable, sterile, single-use device equipped with BLUETOOTH® wireless technology.
- Model 97745 Patient Controller
  The controller is a hand-held device that allows to turn the neurostimulator on and off and check the neurostimulator battery status. It is also used to adjust some of the stimulation settings.
- Model 375003 Boot for Wireless External Neurostimulator
  The external neurostimulator boot is a nonsterile, single-use accessory used to secure the Model 97725 Wireless External Neurostimulator to Participant 's skin with an adhesive pad during trial stimulation.
- Model 97755 Recharger
  The Medtronic Model 97755 Recharger is designed to charge Medtronic rechargeable neurostimulator.
- Model 8880T2 Communicator

The Model 8880T2 Communicator is intended for use by clinicians to use in conjunction with the clinician tablet and clinician programmer app for communication with Medtronic neuromodulation medical devices.

The communicator is handheld and battery-operated. Communication between the communicator and a clinician tablet can occur wirelessly using BLUETOOTH® technology or wired using the USB connector cable.
2- Implanted components for Intellis:
- Model 977D260 Vectris™ 1 × 8 Compact Trial Screening Lead Kit

The Medtronic Vectris 1×8 Compact Model 977D260 Trial Screening Leads is part of a neurostimulator system. The lead has electrodes on the distal end; the proximal (connector) end fits into an

8-conductor connector. A stylet has been inserted into the proximal end of the lead to aid in positioning.

**Spinal segmental mapping.** Following both temporary and permanent implantation, participants were scheduled to perform the process of spinal segmental mapping. Spinal mapping is the process of identifying the right stimulation parameters (frequency, amplitude, and pulse duration) responsible for activation of the lower extremity muscle groups, polarity of the electrodes (cathodes vs. anodes), as well as the number of the channels responsible to evoke the desired contraction and joint movement pattern (hips, knees, and ankles). The mapping protocol may require stimulation for every specific muscle group and joint per each limb.

Spinal mapping was carried out to identify optimal configurations (cathodal-anodal electrode arrangements and stimulation parameters) to enable multiple functions and movements withoutinducing unwanted activity[16,21]. Spinal mapping was conducted daily after temporary (1 week) and permanent (2 weeks) implantation, as well after the first 6 months of the study in an interim mapping phase (4 weeks). Every effort was considered to configure the correct combinations of cathodes and anodes per specific muscle group and joints as well as the correct stimulation parameters for the frequency of the pulses (2–40 Hz) and (250–1000 μs) for the pulse duration. We aimed to use the minimum amount of current (1–10 mA) necessary to evoke muscle contraction and to monitor the increase or decrease during the trial.

From supine position, 12 EMG sensors were attached (left and right vastus medialis, rectus femoris, tibialis anterior, medial hamstring, medial gastrocnemius, and gluteus medius) after shaving and carefully cleaning the skin. The participant was then asked to make three major movements such as wiggling the big toe, dorsiflexion of the ankle joint or moving the entire leg into flexion followed by extension. We aimed to find a multiple electrode combination with the correct stimulation parameters to identify SCES configurations for standing, SCES-evoked recruitment curves from 5 standardized SCES configurations were established based on EMG peak-to-peak amplitude of each muscle group. Recruitment curves were collected by stimulating at 2 Hz from 1–10 mA; three recruitment curves were collected for each configuration at pulse widths of 250, 500, or 1000 μs. From these curves, optimum stimulation configurations that could yield tonic extensor activity for standing were determined. These optimum stimulation configurations were re-tested in a standing frame or standing with a standard walker to establish the standing configurations. Participants were then trained on how to use the SCES controller to activate paralyzed lower extremity muscles.

The emphasis of the first 6 months of the study was to achieve overground standing, while the emphasis of the second 6 months was to achieve step-like activity to lead to overground ambulation. Thus, the initial temporary and permanent mapping periods prior to the first 6 months emphasized voluntary limb movement and extensor activity to facilitate overground standing.

In the interim mapping phase, these original configurations were refined. Configurations which yielded tonic extensor activity in supine were refined to enable standing with the participant upright, first fully supported in a standing frame. Further refining was done by having participants attempt a sit-to-stand in an exoskeleton (EksoNR, Ekso Bionics, CA, USA) in "squat" mode, a manufacturer setting which provides assistive torque at the knees and hips sufficient to complete a sit-to-stand only if the user volitionally contributes to the movement. Lastly, configurations were refined with participants standing in parallel bars. In addition to refinement of configurations for standing, during the interim mapping phase, further mapping was conducted for rhythmic locomotor activity to facilitate stepping and overground ambulation. Different SCES configurations were tested at 2 Hz and 210 μs with incremental amplitudes to see which configurations preferentially recruited the vastus medialis muscles over the medial

gastrocnemius muscles at lower amplitudes[11,14]. Configurations which yielded this activation pattern were then tried at higher frequencies in supine until non-tonic, rhythmic motor output in the lower extremities was found. Configurations which yielded such activity were then tried upright and overground in the exoskeleton, at stimulations amplitudes below levels required to induce rhythmic motor output in the lower extremities in supine. Stimulation amplitudes were then adjusted until participants maximized their step length and steps per minute, and minimized swing phase assistance levels. Step length, steps per minute, and swing phase assistance could all be extracted from training data stored online in the exoskeleton company's proprietary online clinical platform.

**Exoskeleton assisted walking.** Prior to training, a research assistant helped to fit the participant into the device starting with the shoes-support (distally) and then going up toward the trunk (proximally). The software was adjusted and progressed based on the need of each participant. Every effort was made to ensure that all straps were snug but not excessively tight to avoid any episode of autonomic dysreflexia. EAW was scheduled either in the morning or in the evening 3 days/week for the duration of the study. Participants used pro-step+ mode starting with a standard roller walker and then to Canadian Crutches in approximately 4 weeks. The unit is equipped with two buzzers that helps cueing the participants to accurately complete weight shifting prior to stepping[5]. With a trained research assistant providing guidance and support from the back, the participant was encouraged to complete 60-90 minutes fitting, resting and walking time during a single session. Resting and EAW vital signs were monitored in sitting and standing immediately before and after every training session for safety.

**Progression to the adaptability mode.** This mode allows the exoskeleton to gradually lower the assistance provided to the participant based on their performance. In the adaptive assistance mode, the support ranges from 0 to 100%, with 100% means that the unit provides 100% support and assistance to ambulate. During sessions, SCES was turned on and we started with 100% assistance and dropped the support by 5–10% increments each week as tolerated until we reach the lowest assistance level possible during EAW. Decision to drop the assistance will be based on the subject's ability to complete 80% (i.e. arbitrary threshold) of his steps in 10-meter distance without cueing. After dropping the assistance and by selection of a slow mode feature, the exoskeleton offers 2.4 s before fully moving the limb. This period is reasonable for persons with SCI to move their limb in unassisted fashion and if they fail the exoskeleton will beep and finish stepping for them. EAW provides full support at 100%. By dropping the assistance using the adaptive mode, we will be training the subject to volitionally step when we turn the ES-on (i.e. unassisted stepping)[5].

**Overground standing and ambulation without exoskeleton.** Following the exoskeleton session, a follow-up-visit on the same day was conducted to provide over ground standing and walking experience. This started by allowing the subject to stand-up in a standing frame or between parallel bars and to do stepping for 10 feet assisted by a trained researcher. If the subject manages to control standing and stepping between the standing frame or parallel bars, they progressed to perform guarded and supervised walking with a therapist and research assistant for 50 feet using a standard roller walker.

**Measurements**

**Surface EMGs.** After standard skin preparation, surface EMGs were used to record lower extremity and/or trunk muscles, depending on the task being tested. Muscles assessed for mapping and voluntary movement testing included bilateral vastus medialis, rectus femoris, tibialis anterior, hamstring, medial gastrocnemius, and gluteus

medius. Muscles assessed for voluntary trunk and hip extension in semi-standing included bilateral medial hamstrings, vastus lateralis, gluteus medius, transverse abdominis, erector spinae at the level of the T10 vertebrae, and erector spinae at the level of the T5 vertebrae. All EMGs were collected at a 2000 Hz sampling rate using LabChart 8.1.21 (Windows)- ADInstruments. EMGs show in figures are rectified with a 10–990 Hz bandpass filter. In some instances, a root-mean-square filter was layered overtop of the original filtering with a 100-sample window.

**Exoskeleton-assisted walking performance.** Exoskeleton-assisted walking performance variables - steps per minute, step length, and minimum assistance – are derived from data collected by the exoskeleton itself, and then automatically uploaded to the company's online clinical database. Step length and minimum assistance are provided for each individual leg, whereas steps per minute provides the average number of steps per minute taken by both legs for each minute of walking. Step length and minimum assistance values from the left and right legs were averaged to yield one value for each variable, per each step taken by the exoskeleton. Exoskeleton-assisted walking performance values reported represent the average values across 300 steps per session – 5 sessions of exoskeleton-assisted walking values using limb movement SCES are presented next to 5 sessions using rhythmic activity SCES in Fig. 7.

**Overground ambulation.** During post-interventions 1 and 2, overground ambulation without wearing the exoskeleton was tested using parallel bars over a 10-feet (3.05 m) walking distance.

**Peak isometric torque.** This was evaluated using a Biodex isokinetic dynamometer (Shirely, NY). Participants were seated with both the trunk-thigh angle and the knee-thigh angle at 90°. After transferring using a ceiling lift, each participant was securely strapped to the test chair by a crossover shoulder harnesses and a belt across the hip joint. The axis of the dynamometer was aligned to the anatomical knee axis and the lever arm was attached 2-3 cm above the lateral malleolus. Isometric peak torque was measured after implantation, with SCES on and SCES off to allow comparison between the two conditions. Each participant was initially asked to kick his leg as strong as possible to measure the torque generated at the knee joint when the SCES is off and then after 2 minutes resting interval with SCES on. Torque data were collected at a 1000 Hz sampling rate and analyzed using LabChart 8.1.21 (Windows)-ADInstruments.

### Reporting summary

Further information on research design is available in the Nature Portfolio Reporting Summary linked to this article.

## Data availability

De-identified data of the two participants will be available upon direct communication with the corresponding author and upon receiving approval from the appropriate research committee from the Department of Veteran Affairs immediately following publication and available for 3 years. Source data, study protocol and informed consent form are also included with the current submission and limited only for research use. Source data are provided with this paper.

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

## Acknowledgements
The work is currently supported by the DoD-CDMRP clinical trial program award number # W81XWH-20-1-0845 (SC190107 CDMRP W91ZSQ) and Department of Veteran Affairs-SPiRE Program (B3456-P). We would like to thank Center for Rehabilitation Science Engineering Centre-VCU (Ronald Seel, PhD & David Cifu, MD) for providing scientific support to our work. The sponsors did not have any role in study design, data collection and analysis or manuscript writing.

## Author contributions
A.S.G., D.L., and R.T initiated the project and designed the study protocol. L.L.G. and T.D.L. performed clinical assessments. R.T. Intraoperative placement of the SCES and subjects' assessment. A.S.G., T.W.S., J.A.G., and A.A. contributed to data collection, analysis, and interpretation and drafted the manuscript with subsequent contribution from all authors. T.W.S, J.A.G., and A.S.G. supervised all aspects of the work., A.S.G. secured funding

## Competing interests
The authors declare no competing interests.
