## [Peer Review File · Nature Communications]

Interim Report of Percutaneous Epidural Stimulation to Enable Motor Control after Spinal Cord InjuryTransferred manuscripts

This manuscript has been previously reviewed at another journal that is not operating a transparent peer review scheme. This document only contains reviewer comments and rebuttal letters for versions considered at *Nature Communications*.

REVIEWER COMMENTS

Reviewer #1 (Remarks to the Author):

First, I must point out that I do not appreciate the tone taken by the authors when replying to my comments to their original manuscript. I do not have to point out that reviewers put a significant amount of time and effort into providing comments to improve a manuscript. Their responses seem to follow the assumption that I have not carefully read the paper or must have missed—or misunderstood—information in the text. Furthermore, I have pointed out relevant comments regarding the claimed novelty of the work and weaknesses in the methodology, which are still not properly addressed.

(1) In the revised manuscript, the authors have almost completely changed the results section, while leaving out previously provided, yet relevant information (i.e., modulation of induced knee torque in (un)intended direction). This is a critical point since in clinical studies, hypotheses and primary endpoints cannot simply be changed a posteriori.

(2) The manuscript lacks the necessary evidence to support its major claim that SCES had enabled motor control in the two participants with motor complete SCI. Only the supplementary videos 2–4 documented such enabling effects in one participant in lying position.

On the other hand, Figure 1 shows EMG activity induced in multiple lower limb muscles by SCES at 30 Hz with incremental intensities. With lower intensity, tonic EMG activity is induced, which is replaced by rhythmic bursts of activity with increasing intensity – exactly as described 24 years ago with percutaneous epidural leads (Dimitrijevic et al., 1998). There was no enabling of motor control here. The authors claim that the synchronous bursts occurring without reciprocal relationship in antagonistic muscles or left-right alternation led to sudden multijoint flexion movements (not documented by kinematic recordings) that would facilitate overground walking. The walking presented in the supplementary videos 7 and 8 however lacks any functional limb joint flexion.

The modulation in epidurally-induced EMG activity shown in Figures 2 and 3 could merely stem from changes in the proprioceptive input from the legs when changing from sitting to standing, specifically load-receptor related inputs (as previously suggested, e.g. Harkema et al., 2011). The unequivocal demonstration of enabled motor control would require, e.g., under otherwise unchanged conditions (supported standing; constant stim.), the participant to be able to “stop thinking about facilitating the motor task” leading to a collapse of EMG activity. Talking about motor “control”, clearly, the induced activities are not functional, with coactivation of extensors with flexors at multiple joints during the motor task of standing.

Figure 4 shows a comparison of data obtained with SCES using different stimulation parameters, instead of comparing SCES on- and off-conditions. (Also, it is very unusual to show bar diagrams based on only three data points.)

Figure 5 is clearly showing a stance-phase and hence load-related increase in EMG activity across muscles under ongoing stimulation. The strategy of the participant to step is to place one stiff leg and lift the other at the hip, using his upper extremities and residual trunk control, as can be seen in the supplementary videos 7 and 8. Indeed, no relevant EMG activity can be seen in Figure 5 during flexion phases—such activity would however been expected during enabled motor control.

(3) The aspect of electrode migration noticed by the authors in the two participants is a critical methodological flaw, which is considerably toned down in the revision. In participant 0773, there is a migration of ~1.5–2 cm even after permanent implantation (cf. Figure S4) – which by far is not modest as interpreted by the authors. In pain applications of percutaneous systems in specialized clinics, such extent of migration would be a very rare complication. Therefore, the authors should reconsider their

surgical approach and post-surgery care. Regarding the latter, neither percutaneous leads nor surgical leads require 3-4 weeks of immobilization as originally implied by the authors.

Reviewer #2 (Remarks to the Author):

The Authors have strengthened the manuscript by including additional data related to the recovery of standing and stepping.

However, some important aspects of the experimental protocol are missing, and some parts of the manuscript should be improved in their quality and clarity.

1) The legend of Fig. 4 is confusing, especially in relation to the time points in which the experimental data were collected. A 24-week period of EAW Training is mentioned here; however, this part of the experimental protocol is not described anywhere else in the main text or supplemental material.

Similarly, it is unclear after how many training sessions the data related to sit-to-stand, standing, walking with exoskeleton, and walking over-ground, were collected. This is important to contribute interpreting these results.

I strongly suggest that a timeline of data collection be shown in a supplemental figure or table and described in the supplemental methods. The training protocol should be described as well.

2) The EAW-related EMG pattern shown in Fig. 5 should be further commented pointing out its limitations with respect to a 'real' locomotor EMG pattern in which the firing of extensors and flexors is coordinated with respect to the step cycle. This is even more important if EAW is used as a training modality leading to neural plasticity and motor learning.

Minor comments.

- The recovery of overground stepping is probably the most impactful component of this paper, and would deserve dedicated data collection (i.e. EMG) as well as an enhanced interpretation and discussion.

-I could not find the legends of the supplemental videos.

REVIEWER COMMENTS

Reviewer #1 (Remarks to the Author):

First, I must point out that I do not appreciate the tone taken by the authors when replying to my comments to their original manuscript. I do not have to point out that reviewers put a significant amount of time and effort into providing comments to improve a manuscript. Their responses seem to follow the assumption that I have not carefully read the paper or must have missed—or misunderstood—information in the text. Furthermore, I have pointed out relevant comments regarding the claimed novelty of the work and weaknesses in the methodology, which are still not properly addressed.

Answer: We would like to apologize for the reviewer for such tone and we definitely appreciate your time and effort reviewing our report. We did not appreciate the fact that you trying to underestimate the novelty of our work in different directions. We have acknowledged in our revised submission, based solely on your feedback, previous attempts to use percutaneous SCES leads to reduce spasticity and enhance motor recovery in persons with SCI.

We have attempted to highlight the novelty of work in different directions:

1. The use of percutaneous leads with exoskeleton rehabilitation to enhance motor recovery and facilitate restoration of overground ambulation.
2. SCES-enabled sit-to-stand with exoskeleton assistance
3. Exoskeleton-assisted walking performance as far as stepping and walking
4. Induced knee extension torque.
5. overground locomotion using a walker with percutaneous leads

Again, we would like to point out that we are aware there are some limitations to our findings, however we believe that the overall research findings or message from this report override these limitations. Our group is really attempting hard to consider your valuable comments. We have addressed your comments (see below) to improve the manuscript considering the time and effort you have dedicated to our work.

(1) In the revised manuscript, the authors have almost completely changed the results section, while leaving out previously provided, yet relevant information (i.e., modulation of induced knee torque in (un)intended direction). This is a critical point since in clinical studies, hypotheses and primary endpoints cannot simply be changed a posteriori.

Answer: It was not our intention to remove this from our study. The second reviewer pointed out the lack of any functional achievements in the reports. We thought to

prioritize our findings and remove the torque data considering the limited space provided by the Journal for brief reports. We apologize for this and now will relist the torque data in our revised manuscript (lines 140-146; lines 180-181; lines 224-226; and the supplemental table and methods).

(2) The manuscript lacks the necessary evidence to support its major claim that SCES had enabled motor control in the two participants with motor complete SCI. Only the supplementary videos 2–4 documented such enabling effects in one participant in lying position.

Answer: To carefully address this concern. We have to carefully ensure that both of us understand the definition of enabling motor control. Our research group has successfully completed a submission with Dr. Reggie Edgerton and in this work (<https://pubmed.ncbi.nlm.nih.gov/34669485/>), we have carefully defined enabling motor control with epidural stimulation. Enabling motor control means the ability to integrate supraspinal control (spared axonal branches) or proprioceptive Ia or II fibers in the actions of motor unit recruitments to enable motor control, which is now defined in lines 75-77 of the revised manuscript.

As the reviewer clearly pointed that supplemental videos 2-4 demonstrated enabling motor control. However, we cannot rule out based on the definition provided that the other evidence provided in the manuscript demonstrated enabling motor control. Based on additional comments from the reviewer, we have added a figure and supplemental videos that demonstrate that 0772 enabled motor control during hip flexion and extension (see below), and we also believe that he demonstrated enabled motor control when performing a sit-to-stand with exoskeleton assistance (see below).

On the other hand, Figure 1 shows EMG activity induced in multiple lower limb muscles by SCES at 30 Hz with incremental intensities. With lower intensity, tonic EMG activity is induced, which is replaced by rhythmic bursts of activity with increasing intensity – exactly as described 24 years ago with percutaneous epidural leads (Dimitrijevic et al., 1998). There was no enabling of motor control here.

Answer: We have now provided the following figure (figure 3, introduced on line 137) to carefully address your point. When 0773 subject was in side-lying position, we have turned the SCES into 1.3 mA, it is clear in the figure that EMG activity increased into both the right and left legs. We then intentionally asked the subject to move his right leg into flexion. When the subject planned to move his right leg into flexion, EMG activities on the left side decreased remarkably as being demonstrated in the figure. This is considered enabling of motor control; however, we are not in a position to say with our current data whether it is supraspinal control (i.e. volitional intention of the subject) or proprioceptive afferent feedback enhanced by SCES.

Configuration for 0773 resulting in right leg flexion. Left leg muscles are depicted in black, right leg muscles in gray. Both plots occurred simultaneously but have been separated to better visualize different effects of voluntary effort between legs. At 1.3 mA, bilateral muscle activity was induced. However, when the participant voluntarily attempted to flex his right leg, muscle activity decreased in some left leg muscles (VL, RF, HS) while increasing or bursting in right leg muscles (RF, TA, HS, MG). Hz = hertz (stimulation frequency); μ s = microseconds; L = left; R = right; VL = vastus lateralis; RF = rectus femoris; TA = tibialis anterior; HS = hamstrings; MG = medial gastrocnemius; sec = seconds; μ v = microvolts.

The authors claim that the synchronous bursts occurring without reciprocal relationship in antagonistic muscles or left-right alternation led to sudden multijoint flexion movements (not documented by kinematic recordings) that would facilitate overground walking.

Answer: the lack of kinematic recordings in our study is because of failure of access our gait lab due to the COVID-19 restrictions imposed in our center. However, we have clear plans to demonstrate kinematic changes using our VICON system in future implantation. We have addressed the lack of optimal locomotor EMG activity on lines 167-169.

The walking presented in the supplementary videos 7 and 8 however lacks any functional limb joint flexion.

Answer: We are currently refining our mapping protocol to ensure addressing this limitation in future participants. It was extremely challenging to decipher the exact mapping configuration that leads to knee flexion without jeopardizing balance and walking stability during overground ambulation. We have addressed the lack of multi-joint flexion and possible reasons for this in lines 209-215 in the revised manuscript.

The modulation in epidurally-induced EMG activity shown in Figures 2 and 3 could merely stem from changes in the proprioceptive input from the legs when

changing from sitting to standing, specifically load-receptor related inputs (as previously suggested, e.g. Harkema et al., 2011).

Answer: We agree with figure 3 (now figure 6 in the revised manuscript) and we believe that we have carefully addressed this in lines 187-199. However, we believe that figure 2 (now figure 5 in the revised manuscript) is not the result of load-receptor augmentation, but rather shows volitional control. The activity seen in Figure 2 only occurred when the participant volitionally initiated the effort to stand. The exoskeleton provided assistive torque to complete the movement – however, the participant could not move the exoskeleton at all without SCES, as seen in supplemental video 5. Because only a combination of SCES with the participant’s volitional effort yielded both muscle activity and movement, we believe this was not simply due to changes in load-receptor afferent input. This is also discussed in lines 187-199 of the revised manuscript.

The unequivocal demonstration of enabled motor control would require, e.g., under otherwise unchanged conditions (supported standing; constant stim.), the participant to be able to “stop thinking about facilitating the motor task” leading to a collapse of EMG activity. Talking about motor “control”, clearly, the induced activities are not functional, with coactivation of extensors with flexors at multiple joints during the motor task of standing.

Answer: To carefully address your point, we have repeated these tests with subject 0772 and we have listed these findings in the manuscript. This is now shown in figure 4 and supplemental videos 5 and 6, and discussed in the text in lines 156-158 and lines 183-184. We show that the participant was able to control his trunk during volitional hip flexion and extension, and the corresponding increased and decreases in EMG activity which adjusts along with the participant’s volitional intent.

Figure 4 shows a comparison of data obtained with SCES using different stimulation parameters, instead of comparing SCES on- and off-conditions. (Also, it is very unusual to show bar diagrams based on only three data points.)

Answer: We have now revised the entire figure and presented data from the last 5 sessions of limb-movement SCES and data from the first 5 sessions of rhythmic activity SCES. We did not test the SCES off condition at this timepoint due to the study timeline, which is now explained further in the revised Figure 1, introduced on line 76.

Figure 5 is clearly showing a stance-phase and hence load-related increase in EMG activity across muscles under ongoing stimulation. The strategy of the participant to step is to place one stiff leg and lift the other at the hip, using his upper extremities and residual trunk control, as can be seen in the supplementary videos 7 and 8. Indeed, no relevant EMG activity can be seen in Figure 5 during flexion phases—such activity would however be expected during enabled motor control.

Answer: EMG activity presented in Figure 5 (now figure 8 in the revised manuscript)

was during adaptive mode of exoskeleton assisted walking (EAW). We refer to this as atypical firing of EMG. We believe the reviewer has raised an important point. However, we believe a portion of the presented EMG firing is attributed to the exoskeleton-programmed pattern as previously shown by Ramanujam et al (2018). Despite the fact that adaptive mode provides the subjects an opportunity to freely step and to enable motor control, we cannot rule the fact that EAW timing may have interfered with EMG firing presented in figure 5 (now figure 8). Therefore, it is difficult to make any projections between the EMG data presented in figure 5 (now figure 8) and walking performance presented in Videos 7 and 8. We need to point out that the research tools (i.e. EMG recordings) were not accessible in clinical setting environment (videos 7 and 8) because of COVID-19 restrictions that prevented moving equipment from one place to another to reduce likelihood of spreading the virus.

Regarding the other point that was raised by the reviewer that the walking pattern is not enabled by the participant, it is a mix of load receptor-enhanced activation during stance phase and residual trunk movement. We believe that our data initially presented in supplemental videos 2-4 highlighted clearly that the subject enabled motor control. However, at this point our data cannot completely discern between your claim and our claim as a result of lack of real kinematic data that typically would have accompanied our EMG data. We discuss this further in lines 201-208 in the revised manuscript.

(3) The aspect of electrode migration noticed by the authors in the two participants is a critical methodological flaw, which is considerably toned down in the revision. In participant 0773, there is a migration of ~1.5–2 cm even after permanent implantation (cf. Figure S4) – which by far is not modest as interpreted by the authors. In pain applications of percutaneous systems in specialized clinics, such extent of migration would be a very rare complication. Therefore, the authors should reconsider their surgical approach and post-surgery care. Regarding the latter, neither percutaneous leads nor surgical leads require 3-4 weeks of immobilization as originally implied by the authors.

Answer: We totally appreciate your feedback and we acknowledged this clearly in our report that migration in the magnitude of 1.5-2 cm happened in one participant (0773). However, despite the occurrence of migration we have reported that our 0773 subject was able to enable volitional motor control and restoring of overground standing and stepping (videos 7 and 8). The statement regarding immobilization was completely removed in our revised submission. We have addressed the migration concern further in the limitation section in lines 234-239.

Additional comments are also provided to the reviewer regarding the issue of migration in scientific literature:

1. Regarding choice of percutaneous leads over paddles for SC therapy, we realize this is a matter of debate and many centers across the US answer differently. To give a background to our institutional thinking, often cited A study from Duke

neurosurgery department comprised of 13,000+ patients reported at 90 days following the initial procedure, patients in the SCS paddle group were more likely to develop a postoperative complication than patients receiving percutaneous systems (3.4% vs. 2.2%, $p = 0.0005$). (1) This data from 2013 has been repeated with many individual healthcare centers finding similar results and most centers offering either system. I personally perform all SCS percutaneous trials and percutaneous implants for our VA medical center for the last 5 years so the experience is unique. In the case of hardware spanning the target level as in subject 0773, it was deemed that a paddle was not safe for placement.

2. With regard to lead migration, Most recently Dombovy-Johnson et al in 2021 reported on a total of 91 cases (182 leads) Within 20 days of implantation, 88.5% of leads had migrated (86.3% caudal and 2.2% cephalad). Mean migration distance for leads with caudal migration only was 1.234 ± 1.219 cm based on antero-posterior radiographs and 1.695 ± 1.568 cm on lateral radiographs. There was an association of greater caudal lead migration as patient body mass index increased (β -coefficient 0.07 [95% confidence interval 0.01-0.13], $p = 0.031$). This low rate of clinically significant migration which required reoperation is likely attributed to both purposeful cephalad placement and advances in lead programmability. (2)

As astutely pointed out by the reviewer caudal migration did occur in our cases. We would amend the paper, if acceptable, to elaborate on this. We purposely placed the leads a) 2 contacts *higher* than the desired location knowing this. (Medtronic specifically recommends this in both trial and permanent implantation). Additionally, to mitigate lead migration, I b) *sutured to the interspinous ligament* in both cases when the norm for an average patient receiving pain-based therapy, anchoring to lumbar dorsal fascia is considered sufficient. In addition, placing the IPG c) *between the iliac crest and the 12th rib*, ipsilateral to the incision site, ensured that the IPG was in the same anatomical plane as the anchor and entry point regardless of body position, thus reducing lead flexion and mobility. Lastly, we do have the patients relatively immobile for the d) *first 30 days* while epidural scar forms and lead migration becomes significantly lower. In summary, the idea behind offering *modern day* SCS percutaneous lead implantation paired with a percutaneous lead trial with the 4 caveats above has proved successful in our experience for pain-based therapy. Our hope is that we can additionally prove its' utility for motor control and the unique programming requirements. I have, since your thoughtful review, inquired about the titanium mini-plates as an additional adjunct to ensure minimal migration, so thank you for your review.

1. Babu R, Hazzard MA, Huang KT, Ugiliweneza B, Patil CG, Boakye M, et al. Outcomes of percutaneous and paddle lead implantation for spinal cord stimulation: a comparative analysis of complications, reoperation rates, and health-care costs. *Neuromodulation*. 2013;16(5):418-26; discussion 26-7.

2. Dombovy-Johnson ML, D'Souza RS, Thuc Ha C, Hagedorn JM. Incidence and Risk Factors for Spinal Cord Stimulator Lead Migration With or Without Loss of Efficacy: A Retrospective Review of 91 Consecutive Thoracic Lead Implants. *Neuromodulation*. 2021.

Reviewer #2 (Remarks to the Author):

The Authors have strengthened the manuscript by including additional data related to the recovery of standing and stepping. However, some important aspects of the experimental protocol are missing, and some parts of the manuscript should be improved in their quality and clarity.

Answer: We would like to thank the reviewer for his time and effort to read and review our manuscript for the second time. We have attempted every effort to address your concerns,

1) The legend of Fig. 4 is confusing, especially in relation to the time points in which the experimental data were collected. A 24-week period of EAW Training is mentioned here; however, this part of the experimental protocol is not described anywhere else in the main text or supplemental material.

Similarly, it is unclear after how many training sessions the data related to sit-to-stand, standing, walking with exoskeleton, and walking over-ground, were collected. This is important to contribute interpreting these results.

I strongly suggest that a timeline of data collection be shown in a supplemental figure or table and described in the supplemental methods. The training protocol should be described as well.

Answer: We appreciate the reviewer's feedback about figure 4 (now figure 7 in the revised manuscript) legend. Figure 4 (now figure 7) was primarily meant to compare two SCES configurations of mapping during EAW. The findings showed that using the rhythmic SCES configuration elicited better EAW performance.

We totally agree with the reviewer and now we presented a clear timeline for all aspects of the work presented in our manuscript in figure 1. We have originally submitted the work as a brief report and because of the size of the manuscript, we have chosen to focus primarily on the major findings. A clear timeline is now presented that is likely to address the missing aspects of our training protocol in figure 1 and is introduced on line 76.

2) The EAW-related EMG pattern shown in Fig. 5 should be further commented pointing out its limitations with respect to a 'real' locomotor EMG pattern in which the firing of extensors and flexors is coordinated with respect to the step cycle. This is even more important if EAW is used as a training modality leading to neural plasticity and motor learning.

Answer: We totally agree with the reviewer that EMG pattern during EAW may be considered as a limitation to be translated into real coordinated locomotor EMG pattern. This may have presented challenges in translating this pattern of training (i.e. EAW) into over ground standing and stepping (videos 7 and 8). However, in our research center,

the use of EAW is more effective in training participants with SCI and require less involvement from clinicians and research assistants.

Our response to your question, based entirely on figure 5 (now figure 8 in the revised manuscript), when we compared EMG activity during EAW with SCES off versus SCES on, is provided in lines 201-208.

Minor comments.

- The recovery of overground stepping is probably the most impactful component of this paper, and would deserve dedicated data collection (i.e. EMG) as well as an enhanced interpretation and discussion.

Answer: Unfortunately, as result of unanticipated financial crisis and the result of increasing the cost of gas, our 0773 subject has to request withdrawal from the study. We are hopeful that we could bring him back into the study and do further data collection on him. However, additional EMG data collection at this point is not possible.

Because of the COVID-19 restrictions, we failed initially to do any EMG data collection in clinical setting. There are clear restrictions about unnecessarily moving research tools in clinical setting that may likely contribute to additional risks to other patients.

-I could not find the legends of the supplemental videos.

Answer: Thank you, the figure legends are now included.

REVIEWER COMMENTS

Reviewer #1 (Remarks to the Author):

I appreciate the authors' endeavor to promote the use of percutaneous leads as a less invasive method of SCES to facilitate motor function after spinal cord injury. Indeed, SCES has recently gained a resurgence of interest as a potent method to bring about unprecedented levels of motor recovery, and has attracted the attention of individuals with spinal cord injuries and clinical experts alike. While this is an encouraging and exciting development for all those involved, it also comes with increasing responsibility of researchers and scientists in the field to conduct the most rigorous clinical studies possible, in order not to further delay the broader use of SCES as a rehabilitative option—as had happened previously, when the initial positive reports of SCES for motor recovery from the 1970ies were followed by studies in the 1980ies and 1990ies with diverse mythological approaches, some of which lacking a clear hypothesis, describing variant outcomes, and leading to an unfortunate decline in interest in SCES lasting for many years.

Based on this, I unfortunately do feel that the present manuscript lacks the necessary methodological and scientific rigor and also a critical review of existing literature to be considered for publication. The authors' responses to my comments partially raise more questions than have been answered, and in several aspects, the authors contradict themselves. I will list the most important points below:

- In the revised manuscript, the authors have decided to put back the original statement regarding the need for lengthy periods of immobilization after the implantation of epidural paddle electrodes. I am very well familiar with the work of the active groups in the field and can clearly tell that this statement did not apply to any of their studies in individuals with spinal cord injury. The authors use this statement as an argument in favor of the percutaneous leads. To my confusion, the authors stated in response to one of my comments that they had their patients relatively immobile for the first 30 days after implantation of the percutaneous leads – so where was the claimed advantage? There are for sure arguments speaking for percutaneous leads, yet the one about immobilization is simply incorrect. It should also be noted that none of the studies using paddle electrodes have reported any of the complications listed in the introduction of the present manuscript and it seems that the authors aim at drawing an unnecessary negative picture of this procedure.
- The lead migration reported in this manuscript is far beyond that reported in the literature (mean caudal lead migration in the literature: 1.2 ± 1.2 cm). The authors claim that migration in their patients was in the magnitude of 1.5-2 cm, yet, from the legend to the supplementary figure 4, it becomes clear that the distal migration could be up to >4cm, which is simply unacceptable and indicates some amount of improper handling of the methodology. I am glad to hear though that the authors have inquired about additional means to ensure minimal migration for their future implantations.
- The rhythmic, synchronous bursts elicited in the supine position were clearly non-functional and not locomotor-like at all as such activity would have required the presence of a reciprocal relationship between antagonistic muscles or left-right alternations (the authors refer to the synchronous bursts as “not optimal locomotor muscle activity”; this is incorrect). It is unclear how such activity would facilitate overground walking, as claimed by the authors. Accordingly, as I had already stated in my last review, there is no functional limb joint flexion during walking with SCES on (i.e., no enabling of limb flexion).
- Regarding the walking pattern of one of their participants, the videos do not substantiate the authors' claim of “enabled” stepping, but rather stiffened legs (specifically during stance) and a compensatory hip and upper limb strategy to move the legs forward. I am glad to see that the authors agree that they “are not in the position to say with [their] current data whether this is supraspinal control”, as stated in their

response to one of my previous comments.

- The abstract and discussion are partially pretentious regarding the attained effects, implying more motor enabling by SCES than substantiated by their results. For instance, in the abstract, it is briefly mentioned that the subjects could voluntarily modulate SCES-induced knee extension and flexion torques, implying changes in the intended directions, which was, however, not the case. Further, it is claimed that SCES enabled unassisted overground ambulation in one of their patients, thereby implying that no assistive devices were needed, which was, however, not the case.
- The positive effects on orthostatic hypotension in one of the participants is important, yet, lacks any discussion about the potentially underlying mechanisms, the relationship to the specific electrode placement together with the weaker motor effects attained in this patient, and a review of the literature available regarding this specific topic.
- Finally, I do agree with the authors that SCES via percutaneous leads to enable motor control after SCI deserves further exploration and I will gladly see their future publications on this topic once they have gained a better understanding of the methodology, implanted more patients, and collected additional measures that could so far not be obtained, also because of COVID-19 restrictions.
- Minor comment: There is a mismatch between the sequence of the supplementary videos and the corresponding legends.

Reviewer #2 (Remarks to the Author):

The authors included additional data and related interpretations to the manuscript. Overall, the data presented have potential, as they describe progressive and important recovery of motor function for standing and stepping. However, the interpretation of some datasets is flawed and needs to be corrected. Also, because the original brief report has been developed into a full paper, a number of important references that are critical to improve the discussion should be added.

SPECIFIC COMMENTS

P2 line 15. ...self-balance *assistance* (please add "assistance")

P3 line 48. Please modify the part of the sentence as follows: "... persons with *a clinically sensory-motor complete (AIS A) or a motor complete (AIS B)* SCI ..."

Please report the time since injury of the two participants.

P6 line 107. This sentence is confusing because Figure 2 shows data collected only from 773. Please rephrase it and state that exemplary data for 773 are shown in Figure 2.

P6 line 111. "respectively" does not make sense here because only one condition (with stim) is stated in the text. Please revise this sentence.

Torque data. I do have substantial concerns related to the torque data presented by the authors, and their interpretation. (i) Supplemental Fig. 7 shows a torque modulation equal to ~1 Nm during the voluntary effort. This level of torque modulation is physiologically trivial, and it is impossible to know whether it is caused by compensatory movements of the trunk and upper body, which can occur even when the participant is properly strapped on the ergometer, or by activation of key lower limb muscles. Please add this issue in the limitation section. (ii) The data showed in supplemental Table 1 as percent change are completely misleading. Differences of 'a thousand percent' don't make sense when torque levels are negligible as in the present framework. Authors should present the absolute difference of the TTI normalized by time or, even better, the TTI normalized by time of both baseline and effort, so that the reader can understand

the magnitude of torque output generated, and interpret it accordingly.

Figure 4. EMG of trunk muscles with epidural stimulation on cannot be reliably interpreted because a substantial amount of signal detected is electrical noise from the epidural stimulator. The authors should either present additional convincing data to support their view that the trunk EMG traces are not influenced by electrical noise, or remove EMG of trunk muscles.

P8 line 149-150. This sentence reads as Figure 8 shows data from both participants, but it does not. Please rephrase accordingly.

P9 lines 168-170. This sentence and the interpretation of the data in figure 5 is flawed. There is no data showing the effects of proprioceptive inputs alone (i.e. without volitional effort) to be compared to the data presented in figure 5 (volitional + proprioceptive), which would be needed to support the authors' statement. From the studies published on this topic it is very likely that even a passive sit to stand transition would have brought about a robust modulation of the activation patter resulting in facilitation of standing. However, because the authors did not test specifically this condition, they should rephrase the interpretation of figure 5, in that volitional contribution to perform the sit to stand was required to perform this motor task. However, this does not exclude at all the fact that weight-bearing and other proprioceptive information critically contributed to the generation of this activation pattern.

Additionally, studies by Grahn et al., 2017; Rejc & Angeli, 2019; Smith et al., 2022 show and discuss examples of volitional contribution to standing motor output, and can be referenced and briefly discussed in the context of the mechanistic interpretation of these results.

P9 line 173. There are increasing publications related to the mechanisms of improved orthostatic tolerance with scES, and they need to be reported in the discussion, and briefly discussed in relation to the spinal levels targeted by the stimulating percutaneous electrodes, among others.

Percutaneous leads migration is mentioned in the limitations of the study in terms of caudal shifting. However, from the supplemental figures, a medio-lateral shifting can also be observed, with part of the contacts crossing the midline. This is an aspect that can substantially influence side-specific facilitation of epidural stimulation (i.e. Capogrosso et al., 2013) and should be added to this section.

REVIEWER COMMENTS

Reviewer #1 (Remarks to the Author):

I appreciate the authors' endeavor to promote the use of percutaneous leads as a less invasive method of SCES to facilitate motor function after spinal cord injury. Indeed, SCES has recently gained a resurgence of interest as a potent method to bring about unprecedented levels of motor recovery, and has attracted the attention of individuals with spinal cord injuries and clinical experts alike. While this is an encouraging and exciting development for all those involved, it also comes with increasing responsibility of researchers and scientists in the field to conduct the most rigorous clinical studies possible, in order not to further delay the broader use of SCES as a rehabilitative option—as had happened previously, when the initial positive reports of SCES for motor recovery from the 1970ies were followed by studies in the 1980ies and 1990ies with diverse mythological approaches, some of which lacking a clear hypothesis, describing variant outcomes, and leading to an unfortunate decline in interest in SCES lasting for many years.

Response: Thank you for your feedback.

Based on this, I unfortunately do feel that the present manuscript lacks the necessary methodological and scientific rigor and also a critical review of existing literature to be considered for publication. The authors' responses to my comments partially raise more questions than have been answered, and in several aspects, the authors contradict themselves. I will list the most important points below:

Response Supplemental Figures 4 and 5. We would like to thank the reviewer for reading our manuscript now for three times (one time in [redacted] and two submissions in Nature Communication). We, respectfully, want to clarify that our work, similar to other studies in the field, has its own strengths and limitations but not necessarily as the reviewer describes lacks the methodological and scientific rigor.

We have attempted to clarify that when you are referring to the extensive migration > 4 cm that you are referring to **temporary implantation** and not permanent implantation.

To resolve this issue with the reviewer, we have now quantified migration during permanent implantation.

- In 0772, migration during permanent implantation was less than 0.4 cm
- In 0773, migration during permanent implantation was less than 1.6 cm in one of the leads.

According to published reports (see references 13 and 24), longitudinal migration (i.e. overtime) less than 1.7 cm either in the antero-posterior or later views are considered in the acceptable range. Even though, we have clearly addressed this as a limitation and

recommended more aggressive ways to mitigate migration during permanent implantation.

Temporary implantation is only conducted for 5 days to ensure that SCES is successfully working before proceeding to permanent implantation (see below). We believe that migration during permanent implantation is modest and within the published limit.

We hope that the reviewer would balance this with the established novelty of the work rather than focusing on the limitations of the work.

1. Percutaneous SCES enabled standing
2. Percutaneous SCES enhanced exoskeleton performance
3. Percutaneous SCES enabled trunk control in a person with complete cervical injury

We believe that based on the feedback that you provided to us that we have performed a critical review of the published work in the field. You have provided us with important citations similar to the Barlot et al. We would like to acknowledge this and thank the reviewer. Finally, we appreciate your effort to provide critical feedback of our work.

In the revised manuscript, the authors have decided to put back the original statement regarding the need for lengthy periods of immobilization after the implantation of epidural paddle electrodes. I am very well familiar with the work of the active groups in the field and can clearly tell that this statement did not apply to any of their studies in individuals with spinal cord injury.

The authors use this statement as an argument in favor of the percutaneous leads. To my confusion, the authors stated in response to one of my comments that they had their patients relatively immobile for the first 30 days after implantation of the percutaneous leads – so where was the claimed advantage?

There are for sure arguments speaking for percutaneous leads, yet the one about immobilization is simply incorrect.

It should also be noted that none of the studies using paddle electrodes have reported any of the complications listed in the introduction of the present manuscript and it seems that the authors aim at drawing an unnecessary negative picture of this procedure.

Response: We are very sorry but, we have never deleted this from our manuscript.

“I am very well familiar with the work of the active groups in the field and can clearly tell that this statement did not apply to any of their studies in individuals with spinal cord injury

We encourage the reviewer to read the work of Grahn et al. 2017 and examine figure 1. The paper has number of authorities in the field of epidural stimulation. In figure 1 and in

the text, the authors clearly mentioned that there is a clear period of 3 weeks of surgical recovery. This happened before starting any SCES work.

Ref. Grahn PJ, Lavrov IA, Sayenko DG, Van Straaten MG, Gill ML, Strommen JA, Calvert JS, Drubach DI, Beck LA, Linde MB, Thoreson AR, Lopez C, Mendez AA, Gad PN, Gerasimenko YP, Edgerton VR, Zhao KD, Lee KH. Enabling Task-Specific Volitional Motor Functions via Spinal Cord Neuromodulation in a Human With Paraplegia. Mayo Clin Proc. 2017 Apr;92(4):544-554

Response line 33-41: To reconcile with the reviewer and since the reviewer claiming that we are trying to draw negative image about paddle implantation, which is NOT true at all.

We have deleted all references to immobilization period and kept it very simple. We have mentioned that paddle implantation may not be suitable for all patients and therefore, the search for alternative approach may be beneficial to those who suffered from spinal cord injury.

• **The lead migration reported in this manuscript is far beyond that reported in the literature (mean caudal lead migration in the literature: 1.2 ± 1.2 cm). The authors claim that migration in their patients was in the magnitude of 1.5-2 cm, yet, from the legend to the supplementary figure 4, it becomes clear that the distal migration could be up to >4cm, which is simply unacceptable and indicates some amount of improper handling of the methodology. I am glad to hear though that the authors have inquired about additional means to ensure minimal migration for their future implantations.**

Response Supplanted figures 4 and 5 as well as limitation section P12 line 240-250: We have clearly mentioned that temporary implantation was only done *for 5 days* with the stimulator kept outside the body. We have mentioned that the purpose of using temporary implantation is a safe procedure that required by FDA to ensure that epidural stimulation either for pain management or motor control is effectively working before proceeding to permanent implantation.

During permanent implantation, the migration did not exceed (0.4 cm in 0772 and 1.6 cm in 0773 in one of the leads) what has been previously reported in the published trials. We clearly mentioned that we are aware of this limitation, and we are working to mitigate this problem in future implantation. A whole limitation section was written to address this problem.

• **The rhythmic, synchronous bursts elicited in the supine position were clearly non-functional and not locomotor-like at all as such activity would have required the presence of a reciprocal relationship between antagonistic muscles or left-right alternations (the authors refer to the synchronous bursts as “not optimal locomotor muscle activity”; this is incorrect). It is unclear how such activity**

would facilitate overground walking, as claimed by the authors. Accordingly, as I had already stated in my last review, there is no functional limb joint flexion during walking with SCES on (i.e., no enabling of limb flexion).

1. Non-Functional

Response: We, respectfully, disagree with the reviewer. The rhythmic burst enhanced as exoskeleton performance as shown in figure 7. We also believe that rhythmic burst enhanced stepping in the parallel bars and overground with a walker as shown in supplemental videos.

We clearly mentioned that despite clear enabling of the volitional movements, we are not certain whether could be solely from supraspinal control or integrating both supraspinal and proprioceptive inputs together. In you follow-up comment, you seem to be pleased by this addition.

2. Not locomotor

Response P8 line 150156: We did not deny this. You mentioned it is not-locomotor and later you appeared to penalize us when we clearly say this in the manuscript *“configurations for rhythmic activity did not yield optimal locomotor muscle activity in supine position in that the bursting was synchronous across all muscles in both legs”*. We are basically saying the same thing. When we mentioned “did not yield optimal locomotor activity, we meant exactly what you said later in your comment *“lack of a reciprocal relationship between antagonistic muscles or left-right alternations”*. **Please also review figure 8 that clearly showed reciprocal EMG activities between the left and right legs during exoskeleton walking with SCES on.** I think we do not disagree with the reviewer, and we are just confirming his/her observation.

• **Regarding the walking pattern of one of their participants, the videos do not substantiate the authors’ claim of “enabled” stepping, but rather stiffened legs (specifically during stance) and a compensatory hip and upper limb strategy to move the legs forward. I am glad to see that the authors agree that they “are not in the position to say with [their] current data whether this is supraspinal control”, as stated in their response to one of my previous comments.**

Response P11 line 212-218: We believe that his ability to enable flexion patten was overridden by his extensor tone but as clear in the video that patient clearly reciprocate his limbs during different phases of the gait cycle. We have also mentioned that we are in the process of refining our mapping protocol to ensure that we can enable optimal flexion during gait cycle.

• **The abstract and discussion are partially pretentious regarding the attained effects, implying more motor enabling by SCES than substantiated by their results.**

For instance, in the abstract, it is briefly mentioned that the subjects could voluntarily modulate SCES-induced knee extension and flexion torques, implying changes in the intended directions, which was, however, not the case.

Response: Page 2 line 12-14: We apologize for the confusion. Based on the results and based on the supplementary data provided in Table 1, both participants were able to modulate torques induced by SCES after providing verbal commands greater than baseline and with SCES off, but not always in the intended directions. Although 0772 was successfully able to modulate the SCES induced torque in the right direction when using 100% motor threshold at 34 Hz.

Further, it is claimed that SCES enabled unassisted overground ambulation in one of their patients, thereby implying that no assistive devices were needed, which was, however, not the case.

Response: We have clearly clarified this point in the abstract and in the discussion. We have also referred to the use of the parallel bars and walker. Here is what we currently have in the abstract and the discussion

Abstract line 15-16: The same participant achieved independent standing with minimal upper extremity self-balance assistance, independent stepping in parallel bars, overground ambulation with a walker.

- **The positive effects on orthostatic hypotension in one of the participants is important, yet, lacks any discussion about the potentially underlying mechanisms, the relationship to the specific electrode placement together with the weaker motor effects attained in this patient, and a review of the literature available regarding this specific topic.**

Response: We felt that this would result in rather speculative and unnecessary discussion to the text. Since we believe that we do not have the necessary data to back up our statement, we deleted any references to autonomic nervous system from the manuscript.

- **Finally, I do agree with the authors that SCES via percutaneous leads to enable motor control after SCI deserves further exploration and I will gladly see their future publications on this topic once they have gained a better understanding of the methodology, implanted more patients, and collected additional measures that could so far not be obtained, also because of COVID-19 restrictions.**

Response: Thank you, we truly appreciate your critical feedback regarding our work. We cannot deny the fact that your critical review and feedback have significantly improved the quality of the work.

• **Minor comment:** There is a mismatch between the sequence of the supplementary videos and the corresponding legends.

Response Thank you, this has been fixed accordingly.

Reviewer #2 (Remarks to the Author):

The authors included additional data and related interpretations to the manuscript. Overall, the data presented have potential, as they describe progressive and important recovery of motor function for standing and stepping.

Response: Thank you so much for such encouraging and supporting statement to our work. We truly appreciate your scientific feedback.

However, the interpretation of some datasets is flawed and needs to be corrected. Also, because the original brief report has been developed into a full paper, a number of important references that are critical to improve the discussion should be added.

Response: We apologize for any misinterpretation on our behalf, and we truly believe that addition of the recommended references were necessary to improve the quality of the work.

SPECIFIC COMMENTS

P2 line 15. ...self-balance *assistance* (please add “assistance”)

Response line 16: This was added as requested

P3 line 48. Please modify the part of the sentence as follows: “... persons with *a clinically sensory-motor complete (AIS A) or a motor complete (AIS B)* SCI ...”

Response line 50: This was added as requested

Please report the time since injury of the two participants.

Response line 61-62: This was added as requested

P6 line 107. This sentence is confusing because Figure 2 shows data collected only from 773. Please rephrase it and state that exemplary data for 773 are shown in Figure 2.

Response line 111: Thank you, this was clarified as requested.

P6 line 111. “respectively” does not make sense here because only one condition (with stim) is stated in the text. Please revise this sentence.

Response line 116: Thank you, this was deleted.

Torque data. I do have substantial concerns related to the torque data presented by the authors, and their interpretation. (i) Supplemental Fig. 7 shows a torque modulation equal to ~1 Nm during the voluntary effort. This level of torque modulation is physiologically trivial, and it is impossible to know whether it is caused by compensatory movements of the trunk and upper body, which can occur even when the participant is properly strapped on the ergometer, or by activation of key lower limb muscles.

Response line 128-130: We have now revised the entire torque data section and provided clear description on how we measured SCES induced torque time integral and volitional torque time integral with SCES on. Overall, the findings suggested that both participants volitionally modulated the torque above baseline but not in the intended direction.

Please add this issue in the limitation section. (ii) The data showed in supplemental Table 1 as percent change are completely misleading. Differences of 'a thousand percent' don't make sense when torque levels are negligible as in the present framework.

Response line 227-233: Thank you, this was added to limitation section accordingly. We have deleted all the percent changes and presented absolute TTI per your request.

Authors should present the absolute difference of the TTI normalized by time or, even better, the TTI normalized by time of both baseline and effort, so that the reader can understand the magnitude of torque output generated, and interpret it accordingly.

Response in Supplemental Table 1: Thank you, the absolute differences of the TTI normalized by time was added per your request to supplemental table 1.

Figure 4. EMG of trunk muscles with epidural stimulation on cannot be reliably interpreted because a substantial amount of signal detected is electrical noise from the epidural stimulator. The authors should either present additional convincing data to support their view that the trunk EMG traces are not influenced by electrical noise, or remove EMG of trunk muscles.

Response Figure 4: We would like to thank the reviewer for this excellent feedback. We have revised our data analysis and after carefully examining our data, we indeed found electrical noise. We have applied special comb filtering to ensure getting rid of the noise (see reference # 26). We have provided references to the type of filtering that we applied. After applying the filter, it is clear that our participant was capable of enabling his trunk extensors. So, although we noted electrical noise, it is clear that the results did not change.

P8 line 149-150. This sentence reads as Figure 8 shows data from both participants, but it does not. Please rephrase accordingly.

Response line 158: Thank you, this was clarified as requested.

P9 lines 168-170. This sentence and the interpretation of the data in figure 5 is flawed. There is no data showing the effects of proprioceptive inputs alone (i.e. without volitional effort) to be compared to the data presented in figure 5 (volitional + proprioceptive), which would be needed to support the authors' statement. From the studies published on this topic it is very likely that even a passive sit to stand transition would have brought about a robust modulation of the activation patten resulting in facilitation of standing. However, because the authors did not test specifically this condition, they should rephrase the interpretation of figure 5, in that volitional contribution to perform the sit to stand was required to perform this motor task. However, this does not exclude at all the fact that weight-bearing and other proprioceptive information critically contributed to the generation of this activation pattern.

Response line 182-183 as well as 192-202: We have now developed this section and adequately explained that we did not measure or separate proprioceptive inputs from volitional efforts.

Additionally, studies by Grahn et al., 2017; Rejc & Angeli, 2019; Smith et al., 2022 show and discuss examples of volitional contribution to standing motor output, and can be referenced and briefly discussed in the context of the mechanistic interpretation of these results.

Response line 192-202: Thank you so much, we have referenced two of the above studies that helped explaining our result during standing control. We have clearly highlighted your point in the manuscript that both volitional and proprioceptive input have contributed to enabling standing position in one of our patients. We totally agree with the reviewer that we did not test solely whether proprioception contributed to standing or not.

P9 line 173. There are increasing publications related to the mechanisms of improved orthostatic tolerance with scES, and they need to be reported in the discussion, and briefly discussed in relation to the spinal levels targeted by the stimulating percutaneous electrodes, among others.

Response: We agree with the reviewer. However, we felt that this would result in rather speculative and unnecessary discussion to the text. Since we believe that we do not have the necessary data to back up our statement, we deleted the data and any references to autonomic nervous system from the manuscript.

Percutaneous leads migration is mentioned in the limitations of the study in terms of caudal shifting. However, from the supplemental figures, a medio-lateral shifting can also be observed, with part of the contacts crossing the midline. This is an aspect that can substantially influence side-specific facilitation of epidural stimulation (i.e. Capogrosso et al., 2013) and should be added to this section.

Response line 246-251: Thank you, this was clarified as requested and thank you for providing us with the Capogrosso et al., 2013 citation.

REVIEWERS' COMMENTS

Reviewer #1 (Remarks to the Author):

While apparently the authors and this reviewer do not find a common ground regarding some aspects of the manuscript (see the comments of previous reviews), I do appreciate that the revised version as a whole has been largely improved and now also provides the necessary insights into the limitations of the study. The preliminary data derived from two participants may hence provide a useful addition to the existing literature.

- The authors should provide an explanation/hypothesis why two electrode leads were implanted.
- Line 141: The rhythmic EMG bursts shown in Fig 2 are not locomotor muscle activity (this would require e.g. reciprocity between antagonists). Hence, the type of activity is not to be referred to as "not optimal locomotor muscle activity" (it is not locomotor-like, rather than not optimal locomotor-like).
- Line 178: "load receptor augmentation by SCES": please delete "by SCES"

Reviewer #2 (Remarks to the Author):

The Authors improved the previous version of the manuscript.

I still believe that some relevant literature has not been cited, and this is puzzling given the small number of references cited.

Specific comments:

-Paragraph starting at P.6 line 107. Would be important to describe the outcomes of participant 0772 similarly to what has been already done for 0773.

-Supplemental table 1 needs substantial improvements.

1) Please specify in the title that the goal of the task was knee extension.

2) Presenting absolute values improved this table. However, its current version is very confusing. For example, the note in the table states that "negative values indicate extension". SCES induced TTI show some negative values. However, in the "SCES induced direction" column, extension and flexion labels are not in agreement with the positive or negative values reported in SCES induced TTI column.

3) Use same number of decimal digits for both participants.

REVIEWERS' COMMENTS

Reviewer #1 (Remarks to the Author):

While apparently the authors and this reviewer do not find a common ground regarding some aspects of the manuscript (see the comments of previous reviews), I do appreciate that the revised version as a whole has been largely improved and now also provides the necessary insights into the limitations of the study. The preliminary data derived from two participants may hence provide a useful addition to the existing literature.

Answer: We would like to thank the reviewer for his/her excellent feedback regarding our manuscript thorough out the process. We really appreciate your constructive feedback to improve the quality of our work.

- The authors should provide an explanation/hypothesis why two electrode leads were implanted.

Answer line 61-62: Thank you so much. Two leads were implanted to provide multiple configuration options either by adding cathodes or anodes on both leads or on a single lead. This will provide multiple configuration options to enhance functional outcomes.

- Line 141: The rhythmic EMG bursts shown in Fig 2 are not locomotor muscle activity (this would require e.g. reciprocity between antagonists). Hence, the type of activity is not to be referred to as “not optimal locomotor muscle activity” (it is not locomotor-like, rather than not optimal locomotor-like).

Answer line: Thank you so much. Based on your request, we have changed to “...rhythmic activity did not yield locomotor-like muscle activity...”

- Line 178: “load receptor augmentation by SCES”: please delete “by SCES”

Answer: Thank you so much. This was deleted as requested.

Reviewer #2 (Remarks to the Author):

The Authors improved the previous version of the manuscript.

Answer: We would like to thank the reviewer for his time and expertise reviewing our manuscript. We really appreciate your continuous feedback to help improving the quality of our manuscript.

I still believe that some relevant literature has not been cited, and this is puzzling given the small number of references cited.

Answer: We are willing to cite any references that the reviewer would recommend and consider relevant to our report.

Specific comments:

-Paragraph starting at P.6 line 107. Would be important to describe the outcomes of participant 0772 similarly to what has been already done for 0773.

Answer: Thank you so much. We have added the outcome for 0772 based on your request.

-Supplemental table 1 needs substantial improvements.

1) Please specify in the title that the goal of the task was knee extension.

Answer: Thank you so much. We totally agree and this was added to the title of the table.

2) Presenting absolute values improved this table. However, its current version is very confusing. For example, the note in the table states that “negative values indicate extension”. SCES induced TTI show some negative values. However, in the “SCES induced direction” column, extension and flexion labels are not in agreement with the positive or negative values reported in SCES induced TTI column.

Answer: Thank you so much and we apologize for any confusion. We have removed any references to negative or positive values and just kept the direction of the movement.

3) Use same number of decimal digits for both participants.

Answer: Thank you so much. We totally agree and we used the decimal digits to the 10th digit when appropriate for both participants.